# Is It Possible to Forecast the Price of Ḅitcoin?

**Julien Chevallier** [1,2,*,†] , **Dominique Guégan** [3,4,†] **and Stéphane Goutte** [5,6]

1   IPAG Lab, IPAG Business School, 184 Boulevard Saint-Germain, 75006 Paris, France
2   Economics Department, Université Paris 8 (LED), 2 rue de la Liberté, 93526 Saint-Denis, France
3   Applied Mathematics Department, Université Paris 1 Panthéon-Sorbonne, LabEx ReFi,
    106 Boulevard de l'Hopital, CEDEX 13, 75647 Paris, France; dguegan@univ-paris1.fr
4   Department of Economics, University Ca'Foscari of Venezia, 30123 Venice, Italy
5   CEMOTEV, UVSQ, Paris-Saclay, 78280 Guyancourt, France; stephane.goutte@uvsq.fr
6   International School, Vietnam National University, Hanoi 10000, Vietnam
*   Correspondence: julien.chevallier@ipag.fr
†   These authors contributed equally to this work.

**Abstract:** This paper focuses on forecasting the price of Bitcoin, motivated by its market growth and the recent interest of market participants and academics. We deploy six machine learning algorithms (e.g., Artificial Neural Network, Support Vector Machine, Random Forest, *k*-Nearest Neighbours, AdaBoost, Ridge regression), without deciding a priori which one is the 'best' model. The main contribution is to use these data analytics techniques with great caution in the parameterization, instead of classical parametric modelings (AR), to disentangle the non-stationary behavior of the data. As soon as Bitcoin is also used for diversification in portfolios, we need to investigate its interactions with stocks, bonds, foreign exchange, and commodities. We identify that other cryptocurrencies convey enough information to explain the daily variation of Bitcoin's spot and futures prices. Forecasting results point to the segmentation of Bitcoin concerning alternative assets. Finally, trading strategies are implemented.

**Keywords:** forecasting; Bitcoin; machine learning; trading strategies





## 1. Introduction

Artificial intelligence (AI) is the simulation of human intelligence by computers. Misheva et al. [1] underline that AI offers great opportunities for enhancing the customer experience, democratizing financial services, ensure consumer protection and significantly improve risk management. In this field, Bussmann et al. [2] argue that Artificial Intelligence models can be used in credit risk management and, in particular, in measuring the risks that arise when credit is borrowed employing peer to peer lending platforms. Islam et al. [3] recall that a fundamental challenge for A.I.-based prediction models is the extent to which the internal working mechanisms of an AI system can be explained in human terms.

Machine learning (ML) is a branch of AI where algorithms are used to learn from data to make future decisions or predictions. Naturally, forecasting research stands at the forefront of this blooming literature. Cohen [4] uses particle swarm optimization and identifies that both Darvas Box and Linear Regression techniques can help traders predict the bitcoin's price trends. Besides, Li et al. [5] demonstrate that the Attentive LSTM network and an Embedding Network achieve superior state-of-the-art performance among all baselines for the Bitcoin price fluctuation prediction problem. Last but not least, Livieris et al. [6] utilizes as inputs different cryptocurrency data and handles them independently to exploit helpful information from each cryptocurrency separately, which leads to better results than the traditional fully-connected deep neural networks.

At the crossroads between econometrics and machine learning, we find a paper by Chen et al. [7], who assesses that "*the machine learning approach could be a more suitable methodology than traditional statistics for predicting the Bitcoin price*". In this spirit, we attempt

to infer whether such computational methods designed to "learn" information directly from data (and adaptively improve their performance as the number of samples increases) will lead to successful Bitcoin price forecasts. While econometrics assesses potential future scenarios using advanced statistical methods (such as time series), machine learning utilizes artificial intelligence to predict behavior in new ways. Of particular interest to us is that there is still a debate on why a given algorithm can outperform conventional methods in predictive analytics.

Regarding the issue of the use of "black-boxes", there may be fundamental reasons for using them with the suspicion that goes beyond the warnings of Zhao and Hastie [8]. For products that would lend themselves to the use of IA/ML, in practice, these tools are rarely applied for two reasons:

1. Readability: For important investment choices or setting up an investment process, we cannot simply rely on a model. In most cases, it is necessary to have a specific thesis, which must be explained based on simple econometric relations, whether for investors or the CEO of a fund, rarely a specialist. From this point of view, an AR(1) model which would link the BTC to two or three indices could be, from the point of view of investment strategy, more important than a better, more complex model, because it would allow us to better explain and then justify overtime to investors what their money is used for.

2. Far out-of-sample robustness: In setting up complex strategies comparable to derivatives, valuations and measurements of risks are generally based on the simulation of the dynamics of an underlying process (an economic driver of the value of various products). In this case, the input simulations (in addition, calibrated in neutral risk) will generally go well beyond what has been observed in the past, and we have no idea of the relevance of the "black-box" model that will come out. It is then preferable to have a precisely specified model for which the behavior in these out-of-sample areas has been consciously established.

In economics, recent developments of machine learning can be found, for instance, in Farrell et al. [9] who developed semiparametric inference for deep neural networks. In finance, algorithms for quants are already thoroughly documented and accessible, for instance, in the book by de Prado [10].

This paper assesses the risks of machine learning processes as "black-box" (already built) models by detailing precisely the parameters' choices at each step, see also Zhao and Hastie [8], Abadie and Kasy [11]. Adopting a prudent approach towards building sparse models (see the survey on over-fitting and regularization methods by Athey and Imbens [12]), we select six classes of machine learning algorithms: regularization (Ridge regression), neural networks (Multilayer Perceptron with Back-Propagation), ensemble decision trees (Random Forest, AdaBoost), instance-based (*k*-Nearest Neighbour), and classification (Support Vector Machine).

Using these six machine learning processes, we analyze their fitting and predictive power through an empirical application based on Bitcoin spot and futures prices. Along with the paper, we discuss some risks associated with this approach. The paper discusses the potential prediction of the machine learning processes trying to answer the following points:

1. Do drivers exist for Bitcoin forecasts (inside the variables we retain)?
2. Can we accurately produce forecasts (models in question, and how to compare them)?
3. Is it possible to propose robust trading strategies?

Bitcoin is an electronic currency based on a vast peer-to-peer network, totally decentralized. New bitcoins are introduced to the market via a process called mining. The miners receive rewards as soon as they validate recent transactions after solving an optimization problem using a Proof of Work, which needs intensive computation. The first Bitcoin was created in 2009 (Nakamoto [13]). A cryptocurrency can be defined as a digital asset designed to work as a medium of exchange using cryptography to secure the transactions

and to control the creation of additional units of the currency. Since the origin of this cryptocurrency, we have observed high volatility of its price and specific features, which could be an interesting feature to understand, in the context of investment objectives. In February 2021, Bitcoin hit a market capitalization of $1 trillion (all digital coins combined have a market cap of around $1.7 trillion, according to Reuters [14]). By hitting the $1 trillion market cap, the Bitcoin market is gaining acceptance among mainstream investors and companies, from Tesla and Mastercard to the bank BNY Mellon. (See Reuters (2021) at https://www.cnbc.com/2021/02/19/bitcoin-hits-1-trillion-in-market-value-as-cryptocurrency-surge-continues.html, accessed 19 February 2021).

Investment managers are doubtful with respect to the forecastability of Bitcoin, much like currency forecasters. For instance, the Chief Investment Officer of Citi Private Bank, David Bailin, reminds that "*to get an exposure to Bitcoin, if you do not own the actual Bitcoin, any such fund or structure can be a very, very inefficient way to do that*". (See Yahoo Finance (2021) at https://autos.yahoo.com/unstoppable-trends-better-bitcoin-long-142808181.html, accessed on 17 March 2021). Bitcoin's private key custody problem has essentially three practical solutions: (i) kept on an exchange, it constitutes a "honey pot" for hackers (recall the Magic The Gathering Online eXchange (Mt.Gox) where 650,000 BTC were lost); (ii) kept in banks, it undermines the 21 million Bitcoins scarcity by paving the way for securitization all over again (recall the 2008 sub-primes crisis); (iii) kept on a hardware wallet (such as Trezor or Ledger), there is the risk of theft and physical harm (Ledger's marketing database was famously hacked, containing the clients' private addresses). Therefore, no solution appears satisfactory. That is why investment in cryptocurrencies will be a (small) part of the "opportunistic side" of the client's portfolio. According to Amundi's asset managers Vincent Mortier and Didier Borowski, Bitcoin and other cryptocurrencies do not possess the intrinsic qualities of money, i.e., to be a metric unit, a store of value, and a medium of exchange. They do not have any real economic underlying, and there exists no pricing model. Both asset managers are wary of the speculative nature of cryptocurrencies. (See Amundi (2021) at https://research-center.amundi.com/article/crypto-currencies-bubble-or-emergence-new-paradigm-decentralised-finance, accessed on 24 March 2021).

If we investigate the literature on Bitcoin, we observe a considerable amount of papers on this cryptocurrency recently to predict its price or the associated return or to determine the trend of these two quantities. Much literature focuses on the prediction of the volatility for this cryptocurrency. Nearly all the models existing in the linear and non-linear time series have been applied to predict prices or volatility. We give, in the next section, a summary of the more recent papers. Nevertheless, an interesting question remains: is it possible to predict the price of this cryptocurrency, whatever the model used and the period considered? This opens the question of the validity of the conclusions of all these papers. If everything 'works' even if the methods are antinomic, what is the robustness of these predictions?

Motivated by the growth of the Bitcoin market and the recent interest of market participants (for instance, in February 2021 alone, corporate adoption of BTC involved ARK Invest, Blackrock, BNY Mellon, Mastercard, Microstrategy, Square. BlackRock, the world's largest asset manager "started to dabble" in BTC. Tesla invested $1.5B in BTC and announced plans to accept crypto payments. North America's first Bitcoin Exchange-Traded Fund (ETF), the Canadian-based Purpose Bitcoin, amassed $421M in Asset Under Management in its first two days of trading) and academics, this study focuses on machine learning modeling. We illustrate some features that could explain the dynamic behavior of Bitcoin's price by taking into account the non-stationary behavior of the data in place of classical parametric modelings (ARMA, related-GARCH, VAR modelings). For recent extensions in econometrics, see for instance Abedifar et al. [15], Ahelegbey et al. [16] regarding correlation networks, Billio et al. [17] for multivariate models such as Granger Causality, or Baumöhl [18] for connectedness *à la* Diebold and Yilmaz approaches for crypto-assets and exchanges (Dahir et al. [19], Le et al. [20], Mensi et al. [21] and further papers). For recent literature on forecasting non-stationary time series based on machine learning, see,

e.g., Cao and Gu [22], Kurbatsky et al. [23], Wang and Han [24]. The interest of machine learning is that the notion of non-stationarity is not crucial as in econometric models for which we need to have stationarity to be sure to have a solution (because this corresponds to the assumptions imposed by econometric models), which is irrelevant in ML. In that sense, machine learning could be a promising technology. We know that it has been used and advanced for asset price/return prediction in recent years since the financial time series are non-stationary and volatile. The development of machine learning and its interest in finance is not new. A seminal paper on the introduction of this methodology was given by Rosenblatt [25], and more recent developments can be found in Russell and Norvig [26] with a lot of references therein. For applications using financial assets, the current paper of Iworiso and Vrontos [27] provides evidence that machine learning techniques permit us to get exciting results concerning the forecasts of the direction of the U.S. equity premium.

In the spirit of the previously-cited papers, in the present paper, we analyze the behavior of the Bitcoin cryptocurrency and its futures with a class of machine learning techniques. We investigate its behavior in the future. As soon as Bitcoin is used for diversification in portfolios, we complete our analysis by looking at Bitcoin's interaction with stocks, commodities, bonds, and other cryptocurrencies. Our research based on these data analytics techniques focuses on (i) their capability to fit a data set. We observe that the Adaboost method and the random forest processes are the winners inside a competition based on six competitors. (ii) Regarding their predictability power, we observe a high variability of the results depending on the period on which we work and the input data used for the training. Thus, the question that emerges from this work is the possibility of predicting the spot or the future for this cryptocurrency against luck or uncertainty, without obviously calling into question the methodology used.

Central banks convey this idea of Bitcoin being an extremely inefficient way to process transactions, highly speculative, and used mainly for the financing of illicit activities. In the view of the Treasury Secretary Ms. Janet Yellen, Central Banks Digital Currencies (CBDC) should be the only solution for printing digital money (through its proprietary core ledger). This view is largely echoed in finance journals. To cite a few, Foley et al. [28] estimate that around $76 billion worth of illegal activity per year involve Bitcoin (46% of bitcoin transactions), which is close to the scale of the U.S. and European markets for illegal drugs. Among other "Silk Roads" dismantled by the FBI regarding drug trafficking, the risks of "black e-commerce" are heightened by the anonymous file server Tor (The Onion Router), and by secret cryptocurrencies' operational design such as Zcash or Monero.

Spanning daily data from 13 January 2015, to 31 December 2020, our analysis is based on several steps to analyze the main drivers of the Bitcoin currency. First, we look at the realm of seventeen cryptocurrencies. Second, as representative of traditional financial markets, we investigate the relationships of Bitcoin with eleven stocks, four bonds, and four foreign exchange markets. Third, we examine the interactions with four energy, seven metals, three grain commodities, five softs, and two cattles as an alternative investment class. The analysis is robust to Bitcoin spot or futures prices as the underlying asset. The novelty lies in (i) considering six machine learning models and one parametric model (an AR) in a horse race to forecast the price of Bitcoin, (ii) developing trading strategies issues to investigate the potential use of crypto assets in portfolio management. As robustness checks, we identify several sub-sample forecasts for results sensitivity purposes.

Regarding the central methodological and empirical contributions, our paper stresses the key ingredients to make a 'good' machine learning model in quantitative economics, a.k.a: (i) proceeding to an excellent data collection (our 'financial markets' approach as opposed to a 'blockchain approach' feeding the models with technical and non-stationary data) while controlling for low multi-collinearities; (ii) assessing using a wide array of visualization tools (clusters, maps, diagrams) the main finding of segmentation of Bitcoin concerning traditional financial and commodity markets (e.g., Bitcoin reacts mainly to the information content of other cryptocurrencies); and (iii) favoring either the AdaBoost or

Random forest algorithms as predictors of the Bitcoin spot and futures prices, which allows us to implement trading strategies; and to open the debate on the forecasting accuracy of Bitcoin.

What is the accurate information set $\{I\}$ to predict Bitcoin prices? Our reply is largely that Bitcoin appears segmented to crypto-assets only, and not much connected to financial markets. Therefore, we do not deploy standard econometrics tests (with Granger causality or reverse causality). We follow a purely data-driven machine learning approach. In a nutshell, this paper contains the results of a set of prediction exercises. The critical emphasis is placed on the proper use of machine learning techniques (Artificial Neural Network, Support Vector Machine, Random Forest, k-Nearest Neighbours, AdaBoost, Ridge regression) to forecast daily movements of the price of Bitcoin. We demonstrate that the performance of such machine learning methods is highly dependent on several design choices (hyperparameters, optimizers, network topology). The forecast statistics retained are the Root Mean Square Error (RMSE), the Mean Absolute Error (MAE), and the Mean Absolute Percent, Error (MAPE). The paper concludes that, in this particular exercise, AdaBoost stands out as the best machine learning. The Random Forest algorithm also performs well among the six considered.

The remainder of the paper is organized as follows. Section 2 summarizes some of the papers interested in predicting Bitcoin with different modelings, classifying these modelings concerning the models. Section 3 describes, in a uniform way, all the machine learning models used. Section 4 introduces the data. Section 5 contains the results for the whole sample, distinguishing the results obtained with all the risk factors we have listed to explain the spot's behavior and Bitcoin futures. This section proposes an in-depth analysis of the inter-relationships between Bitcoin, other cryptocurrencies, and the stable coin Tether, traditional asset markets, and alternative commodities. Section 6 provides robustness checks along four sub-samples corresponding to different periods characterizing Bitcoin's price behavior: a restricted sample to the newest cryptocurrencies, Tether's introduction in 2017, the 2016–2018 Bitcoin economic cycle, the recent 2019 trend, and the 2020 "bull run". Section 7 provides some discussions and conclusions.

## 2. Background

The literature on Bitcoin pricing is developing in finance. Among various topics tackled, Easley et al. [29] document the level of transaction fees on this particular market and assess that a high volume of transactions is required. Bitcoin mining is computationally intensive on the network, and a model calibration was achieved by Prat and Walter [30] (including the electricity cost). Mining rewards are the main incentives for miners to invest in expensive mining pieces of equipment (e.g., dedicated GPU cards or ASIC miners). Hence, the motto "Get Rich or Die Mining" is often found on crypto forums). Another concern on this market is the ability of traders to benefit from price deviations that occur due to multiple trading places: this is called arbitrage between exchanges (Makarov and Schoar [31]). In management, the focus is more on the "cryptocurrency mania" that risks leading to speculative bubbles, as in Cheng et al. [32], Wei and Dukes [33]. Financial practitioners are also concerned about the security of the blockchain (Pagnotta [34]). Quantum computers are posing a serious challenge to the security of the Bitcoin blockchain indeed. (See Deloitte (2021) at https://www2.deloitte.com/nl/nl/pages/innovatie/artikelen/quantum-computers-and-the-bitcoin-blockchain.html, accessed on 14 March 2021).

Several studies have been conducted in the literature concerning predicting Bitcoin spot price or the evolution of its volatility trend. We provide some references without being exhaustive.

Some are based on classical econometric modeling, including: (i) time-series techniques (e.g., vector autoregressive (VAR), vector error correction (VEC), quantile regression), for instance, see Fantazzini et al. [35] and references therein. (ii) GARCH and DCC modeling: for example, Briere et al. [36] investigate the volatility behavior of Bitcoin. Using the same models, Aslanidis et al. [37] compare the volatility of different cryptocurren-

cies, including Monero. Caporale and Zekokh [38] use Markov switching modelings to investigate the volatility of Bitcoin and other cryptocurrencies. (iii) Long memory and jump modeling: in several papers, authors try to detect a possible long memory behavior using different techniques, see, for instance, Bariviera et al. [39], Alvarez-Ramirez et al. [40], Begušić et al. [41]. Some authors try to use this long memory behavior for trading strategies; see, for instance, Khuntia and Pattanayak [42], Al-Yahyaee et al. [43]. With the possible long memory behavior, models with jumps have been used to investigate both the returns' behavior and the volatility. Some references are Phillip et al. [44], Mensi et al. [45]. There exists a large literature on the bubble behavior of Bitcoin, which has been observed since 2014. We can cite, among others, Su et al. [46], Guegan and Frunza [47], Geuder et al. [48]. At the same time, looking at the evolution of the price on specific periods, authors try to show that Bitcoin can be considered a commodity (the idea is that this cryptocurrency corresponds to a limited resource), e.g., Guesmi et al. [49], or as gold (Dyhrberg [50]). In many cases, the authors are interested in discussing the potential (or not) of Bitcoin for diversification, (e.g., Polasik et al. [51], Bouri et al. [52], Selmi et al. [53]).

Some papers use high-frequency data and are interested in shock transmission: using realized volatility of the cryptocurrencies, some authors detect asymmetries in shock transmissions between the cryptocurrencies and traditional assets, see, for instance, Kurka [54] and references therein. The informational efficiency of Bitcoin has also been investigated using high-frequency in Zargar and Kumar [55], extending some previous works on different papers whose references can be found in this last paper.

Some papers have investigated the cross-correlation between cryptocurrencies and different stocks and bonds using related GARCH and DCC modelings to use Bitcoin for diversification. For instance, in a recent paper, Aslanidis et al. [37] detect that the correlation of traditional assets against Monero is even closer to zero than against other cryptocurrencies. Other papers investigate the correlation with different stocks, such as Fang et al. [56], Gillaizeau et al. [57], among others.

Sentiment analysis using Twitter and Google Trends forms another new tool to forecast Bitcoin prices. For instance, Wołk [58] recently mobilized this computational tool to predict the prices of Bitcoin and other cryptocurrencies for different time intervals. The author highlights that people's psychological and behavioral attitudes significantly impact the highly speculative cryptocurrency prices. Further, on informative signals derived from Twitter and Google Trends, Shen et al. [59] find that the number of tweets is a significant driver of next-day trading Bitcoin volume. Philippas et al. [60] identify that Bitcoin prices are partially driven by momentum on media attention in social networks, justifying a sentimental appetite for information demand. Guégan and Renault [61] explore the relationship between investor sentiment on social media and intraday Bitcoin returns. The authors document a statistically significant relationship between investor sentiment and Bitcoin returns for frequencies of up to 15 min. The impact of news is further documented by Dey et al. [62] regarding the use of chainlets to evaluate the role of the local topological structure of the blockchain on the joint Bitcoin and Litecoin price formation and dynamics, or by Nicola et al. [63] regarding information theory measures extracted from a Gaussian Graphical Model constructed from daily stock time series of listed US banks.

Finally, machine learning modeling has recently been used to understand the behavior of cryptocurrencies. Atsalakis et al. [64], Jang and Lee [65], Mallqui and Fernandes [66] investigate the direction prices for daily cryptocurrencies. Atsalakis et al. [64] uses a hybrid Neuro-Fuzzy controller based on artificial neural networks for Bitcoin prices. Jang and Lee [65], for the same data set, use a Bayesian neural network. Mallqui and Fernandes [66] focus on Artificial Neural Networks (ANN), Support Vector Machines (SVM), and k-Means clustering method for Bitcoin predictions introducing other stocks in their study. On another side, Nakano et al. [67] explore Bitcoin intraday technical trading strategies based on deep learning for the price direction return prediction (up and down) on the period of December 2017 January 2018. They provide interesting results on the role of the layers, outputs, and inputs for their trading strategies. Sun et al. [68] adopt a novel Gradient Boosting

Decision Tree (GBDT) algorithm, Light Gradient Boosting Machine (LightGBM), to forecast the price trend. Further on this, [69] hierarchically cluster Bitcoin prices from different exchanges and classic assets by enriching the correlation-based minimum spanning tree method with a primary filtering method based on the random matrix approach. Using a stochastic neural network model, Jay et al. [70] trained the Multi-Layer Perceptron (MLP) and Long Short-Term Memory (LSTM) models for Bitcoin, Ethereum, and Litecoin. The results show that the proposed model is superior in comparison to the deterministic models.

Our paper is close to this last class of articles, with new and different findings.

### 3. Methodology

This section explains the learning algorithms we used. We aimed to build models that make predictions based on a known set of input data. We trained the models to generate accurate predictions when including new data.

In what follows, we describe training the different models using data. We introduce a general formalism permitting applying the models without going into details and providing specific references for more details. Indeed all these models are well-documented in the literature. As soon as we compare several non-parametric modelings, we uniformly present them to compare the training on the data set more accessible. In the next section, we will specify the values of the parameters that have been chosen to provide better forecasts for each model. In what follows, we deploy six "off-the-shelf" ML algorithms that vary depending on the speed of training, memory usage, predictive accuracy, and interpretability.

*'Horse Race' of Machine Learning Models*

Against the benchmark AR(1) parametric model (a.k.a, the standard workhorse of time series econometrics), we retained six non-parametric models: the Ridge/Lasso regression, which can be used as a benchmark, an artificial neural network, a random forest modeling, a support vector machine, the *k*-nearest neighbors approach and the Ada-boost modeling. All these modelings can be associated with a regression based on input factors *X*, providing an output $\hat{Y}$, which is the forecast we expect. Thus formally, we have the following representation: $\hat{Y} = f(X)$, and *Y* is the unknown true objective to attain. The regression function *f* will be more or less complicated, depending on the model we consider.

We provide in Figure 1, a general representation of the framework we used. In the following, we specify the target function *f* for each modeling and the fitting parameters.

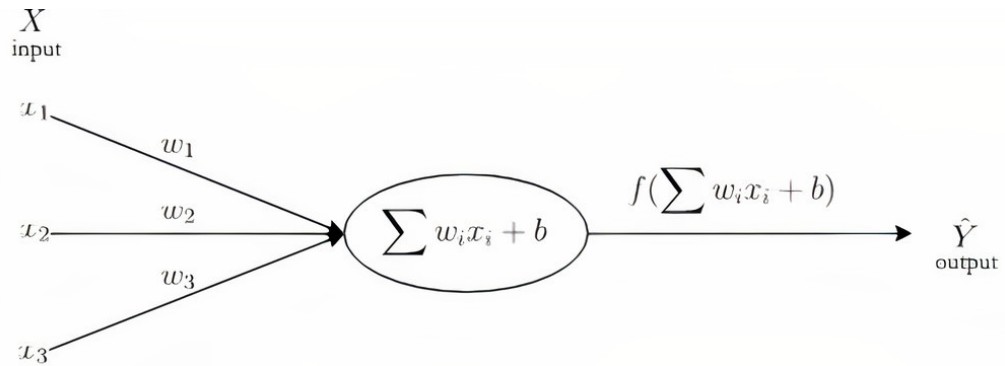

**Figure 1.** Representation of the non parametric modelling used in this paper.

Linear regression is an example of a parametric approach assuming a linear functional form for $f(.)$. Parametric methods have several advantages. They are often easy to fit because one needs to estimate only a small number of coefficients. In linear regression, the coefficients have simple interpretations, and statistical significance tests can be easily performed. However, parametric methods have a disadvantage: by construction, they make strong assumptions about the form of $f(.)$. If the specified functional form is far from the truth, and prediction accuracy is our goal, then the parametric method will perform poorly. In contrast, non-parametric methods do not explicitly assume a parametric form

for $f(.)$ and thereby provide an alternative and more flexible approach for performing regression. We propose various non-parametric methods in this paper.

1. Linear regression. Denoting $Y$ the output and $X$ the centered and standardized inputs, and considering a data set $(y_i, x_i)$, $i = 1, \cdots, n$, the elastic net regression approach solves the following problem

$$min_{\beta_0, \beta} \left[ \frac{1}{2N} \sum_{i=1}^{n} (y_i - f(x_i))^2 + \lambda P_\alpha(\beta) \right], \tag{1}$$

where $f(x_i) = \beta_0 - x_i^T \beta$ (here $T$ is used for transpose), and the elastic net penalty is determined by the value of $\alpha$:

$$P_\alpha(\beta) = (1 - \alpha) \frac{1}{2} \|\beta\|_{l_2}^2 + \alpha \|\beta\|_{l_1}. \tag{2}$$

This elastic-net penalty term is a compromise between the Ridge regression ($\alpha = 0$) and the Lasso penalty ($\alpha = 1$): the constraint for minimization is that $P_\alpha(\beta) < t$ for some $t$. Historically, this method has been developed when the number of variables $p$ is vast comparing to $n$, the sample size. The Ridge method is known to shrink the correlated predictors' coefficients towards each other, borrowing strength from each other. Ridge regression typically fits a model that can predict the probability of a binary response to one class or the other. Lasso is indifferent to correlated predictors. Thus, the role of $\alpha$ is determinant: in presence of correlation, we expect $\alpha$ close to 1 ($\alpha = 1 - \varepsilon$, for small $\varepsilon$). It also exists some link between $\lambda$ and $\alpha$. Generally, a grid is considered for $\lambda$ as soon as $\alpha$ is fixed. A $l_q$ ($1 < q < 2$) penalty term could be also considered for prediction. The regularization done with this penalty term permits to avoid over-fitting.

The algorithm also proposes a way to update the computation, optimizing the number of operations needed. It is possible to associate a weight $\beta_i, i = 1, \ldots, n$, to each observation, which does not increase the computational cost of the algorithm as long as the weights remain fixed. In the following, we use linear regression. Thus the response belongs to $R$. The parameter of interest is $\alpha$, other parameters to estimate are $\lambda, \beta_i, i = 1, \ldots, n$. The existence of correlation must be taken into account to verify whether the values used for those parameters are efficient. For estimation, the parameter $\alpha$ has to be chosen first. Simple least-squares estimates are used for linear regression, but a soft threshold is introduced to consider the penalty term through the decrementation of the parameter $\lambda$ using loops.

The Lasso representation is described in Tibshirani [71], and the elastic net approach is developed in Zou and Hastie [72]. Other recent and interesting references are Fan and Li [73], Zou [74], Zhao and Yu [75], Hastie et al. [76], Tibshirani [77] and Epprecht et al. [78].

2. Artificial Neural Network (ANN). Inspired by the human brain, a neural network consists of highly interconnected neurons that relate the inputs to the desired outputs. The network is trained by iteratively modifying the connections' strengths to map the given inputs to the correct response. ANNs are best-used for modeling highly nonlinear systems, when the data is available incrementally, and when there could be expected changes in the input data. Supervised ANNs were essentially variants of linear regression methods. A standard neural network consists of many simple, connected processors called neurons, producing a sequence of real-valued activations. Input neurons get activated through sensors perceiving the environment. Other neurons get activated through weighted connections from previously active neurons. An efficient gradient descent method for teacher-based supervised learning in discrete, differentiable networks of arbitrary depth called back-propagation is used to attain the algorithm's convergence. This paper uses stochastic gradient descent, a stochastic

approximation of the gradient descent optimization, and an iterative method for minimizing the objective function $f$ written as a sum of differentiable functions. The classical neural network layer performs a convolution on a given sequence X, outputting another sequence Y whose value at time t is:

$$\hat{Y} = \sum_{j=1}^{p} f(\beta_j, x_j(t)) + \varepsilon(t), \tag{3}$$

where $\beta_j$ are the parameters of the layer trained by back-propagation. The parameters to choose are the number of layers and the stopping criteria for convergence purposes. In this paper, we consider an Artificial Neural Network. The ANN is an algorithm that allows for drawing more complex patterns and relationships. Training an ANN to make predictions using back-propagation requires iterating over a two-step process described as follows. (i) We computed the predictions using the previous weights, also known as a forward process; for the first iteration, the weights are often initialized randomly to prevent symmetry issues. (ii) We calculated the gradients to amend the weights for the next iteration, using the same weights as in step one and the freshly computed prediction. The stochastic gradient descent uses a penalty term (for regularization) based on the derivatives making computational the method for finding the approximate optimum and convergence slow. This process generally leads to a local optimum (instead of a global optimum), which would minimize the mean squared errors between the estimated and valid values only locally.

Thus, in more detail, an ANN is a structure of multiple layers, themselves composed of several units, known as neurons of the form, at a step $j$ figures:

$$\hat{Y} = f(\beta_j + \sum_{i=1}^{n} \beta_{ij} x_i), \tag{4}$$

where $f(.)$ is a non-linear activation function, basically the sigmoid function. Assuming the same function is used for the whole structure, it is then used recursively throughout the neural network, inputting each previously computed function $f(.)$ into the next layer's neurons. A description step by step yields clarity to this black-box structure: (i) a first layer gathers the raw data, thus representing the model's inputs $X = (x_1, x_2, \ldots, x_n)$. It is composed of as many neurons as there are samples, each containing $X_i = (x_{i1}, x_{i2}, \ldots, x_{im})^T$. These are forwarded to the next (hidden) layer's neurons via the synapses; (ii) a second layer follows, called the hidden layer (since no true visibility is gained on the meaning of its calculations). Each of its neurons computes a weighted average of all the previous layers' output and incorporates it as $X_i$ in its activation function. Then, it, in turn, forwards the computed value to the next layer; (iii) the last layer, called the output layer, finally computes a weighted average of the hidden layer's neurons outputs and produces a prediction $\hat{Y}$.

Details on neural networks can be found in Maclin et al. [79], Vapnik [80] and Scholkopf [81]. Recent references are Windisch [82] and Hinton and Salakhutdinov [83]. A review paper is the one by Schmidhuber [84]. Note that in the present paper, we do not use deep learning modelings as soon as the set of data we consider is not sufficiently large to justify the expectation of having good results with this sophisticated method, which is appropriate for a huge amount of data and specific data sets.

3. Random forests. Random forest is an ensemble learning method used for classification. Ho [85] first proposed it. Breiman [86] further developed it. Random forest builds a set of decision trees. Each tree is developed from a bootstrap sample from the training data. When developing individual trees, an arbitrary subset of attributes is drawn (hence the term 'random'), from which the best attribute for the split is selected. The number of branches and the values of weights are determined in the training process. The final model is based on the majority vote from individually developed trees in the forest.

An additive tree model (ATM) is an ensemble of $p$ decision trees. Let $X$ be the vector of individual features. Each decision tree outputs a real value. Let $f_p(x)$ be the output from tree $p$. For both classification and regression purpose, the output $f$ of the additive tree model is a weighted sum of all the tree outputs as follows:

$$f(x) = \sum_{j=1}^{p} \omega_j f_j(x), \tag{5}$$

where $\omega_j \in \mathbb{R}$ is the weight associated to tree $j$.

The previous formulation is very general and includes some popular models as special cases, like random forests. This additive tree model is widely used in real-world applications and appears to be the most popular and influential off-the-shelf classifier. It can cope with regression and multi-class classification on both categorical and numerical datasets with superior accuracy. In these ensemble methods, several weaker decision trees are combined into a more robust ensemble. A bagged decision tree consists of trees trained independently on data that is bootstrapped from the input data. In essence, the random forest is a bagging model ([87]) of trees where each tree is trained independently on a group of randomly sampled instances with randomly selected features.

The random forest consists in combining the $p$ regression-type predictors $r_j$ to build another predictor

$$r_\omega(x) = \sum_{j=1}^{p} \beta_j r_j(x),$$

for every $x \in X$. The vector of weights $\beta = (\beta_1, \ldots, \beta_M)$ has to be chosen carefully. Even if the weights could depend on $x$, we keep them constant for simplicity. In the usual case, they are all equal to $1/p$, even if some attempts have been made to add another degree of flexibility with different weights.

If $X$ have uniform distribution on $[0,1]^d$, then the response of the modeling is

$$\hat{Y} = \sum_{j \in S} \beta_j x_j + \varepsilon, \tag{6}$$

where $S$ is a non-empty subset of $d$ features. We chose the following parameters with this modeling: the number of trees and the stopping criteria used to choose among the most significant variables. Depending on the context and the selection procedure, the informative probability $p_j \in (0,1)$ may obey certain constraints positiveness and $\sum_{j \in S} p_j = 1$. It is well-known that for randomized methods, the behavior of prediction error is a monotonically decreasing function of $p$, so in principle, the higher the value of M, the better from the accuracy point of view.

Thus, the question is how to introduce flexibility in the regression functions used in regression trees and their extension to random forests. One splits the sample into sub-samples and estimates the regression function within the sub-samples simply as the average outcome. The splits are sequential and based on a single co-variate $X_{ik}, \{i = 1, \ldots, n\}, \{k = 1, \ldots, p\}$ at a time exceeding a threshold $c$. The outcomes are provided minimizing the average squared error over all co-variates $k$ and all thresholds $c$, then repeating this over the sub-samples and leaves: At each split,

the average squared error is further reduced (or stays the same). Therefore, we need regularization to avoid the over-fitting that would result from splitting the sample too often. One approach is to add a penalty term to the sum of squared residuals linear in the number of sub-samples (the leaves). The coefficient on this penalty term is then chosen through cross-validation.

Random forests has been proposed by Breiman [86] for building a predictor ensemble with a set of decision trees that grow in randomly selected sub-spaces of data, see also Geurts et al. [88] or Biau [89], and for a review, Genuer et al. [90]. The bagging approach is due to Breiman [87]. The discussion on the choice of the weights was done by Maudes et al. [91].

4. Support Vector Machines (SVM). SVM map inputs to higher-dimensional feature spaces. It has been introduced within the context of statistical learning theory and structural risk minimization. The SVM classifies data by finding the linear decision boundary (e.g., hyperplane) that separates all data points of one class from those of the other class. This machine-learning algorithm separates the attribute space with a hyperplane, maximizing the margin between the instances of different classes or class values. It can be used when the researcher needs a classifier that is simple, easy to interpret, and accurate.

If we consider SVM from a regression approach, it performs linear regression in a high-dimension feature using a $\epsilon$- insensitive loss. Its estimation accuracy depends on a suitable setting of the different parameters. The SVM map inputs $X$ to higher-dimensional feature spaces. The support vector machine accommodates nonlinear class boundaries. It is intended for the binary classification setting in which there are two classes. The basic idea is to divide a $p$-dimensional space (called hyperplane) into two halves. In dimension two, a hyperplane is a line.

Considering a data set $(y_i, x_i)$, $i = 1, \ldots, n$, The linear support vector classifier can be represented as

$$f(x) = \beta_0 + \sum_{i=1}^{n} \beta_i \sum_{j=1}^{p} x_{ij} y_{ij}. \tag{7}$$

To estimate the parameters $\beta_i$, all we need are the $\frac{n(n-1)}{2}$ products $x_i y_i$ between all pairs of training observations, where $x_i = \sum_{j=1}^{p} x_{ij}$ and $y_i = \sum_{j=1}^{p} y_{ij}$. So, if $S$ is the collection of indices of these support points $x_i, y_i$, we can rewrite any solution function of the previous form as

$$f(x) = \beta_0 + \sum_{i \in S} \beta_i x_i y_i. \tag{8}$$

Note that a more general representation of the nonlinear function has the form

$$f(x) = \beta_0 + \sum_{i \in S} K(x_i, y_i), \tag{9}$$

where $K(.,.)$ is some function that we will refer to as a kernel. A kernel is a function that quantifies the similarity between two observations.

Some references for the details on support vector machines are Friedman et al. [92] and James et al. [93].

5. *k*-Nearest-Neighbors (*k*-NN). *k*-NN categorizes objects based on the classes of their nearest neighbors in the dataset. Distance metrics are used to find the nearest neighbor. The *k*-NN algorithm searches for *k* closest training instances in the feature space and uses their average prediction. *k*-NN predictions assume that objects near each other are similar. When mobilizing *k*-NNs, memory usage and prediction speed of the trained model are of lesser concern to the modeler.

The *k*-NN regression method is probably the simplest non-parametric method we can propose. It works as follows: given a value for *k* and a prediction point of $x_0$, *k*-NN regression first identifies the *k* training observations closest to $x_0$, represented by $S_0$.

Then it estimates $f(x_0)$ using the average of all the training responses in $S_0$. In other words, we get

$$f(x_0) = \sum_{x_i \in S_0} \beta_i x_i. \tag{10}$$

In general, the optimal value for $k$ will depend on the bias–variance trade-off. A small value for $k$ provides the most flexible fit, which will have low bias but high variance (because the prediction, in that case, can be entirely dependent on just one observation). In contrast, larger values of $k$ provide a smoother and less variable fit; the prediction in a region is an average of several points. Changing one observation has a more negligible effect.

Using this method, we need to estimate the parameter $k$ and decide the weights $\beta$ associated with each point. We often use uniform weight $\beta_k = \frac{1}{k}$: all points in each neighborhood are weighted equally. It is also important to note that closer neighbors of a query point have a more substantial influence than the neighbors further away. Some references on nearest neighbors are Friedman et al. [94], Dasarathy and Belur [95], and, more recently, Papadopoulos and Manolopoulos [96]. Carrying out of the method can be found for instance in Sorjamaa et al. [97].

6. Ada-boosting. In these methods, several "weaker" decision trees are combined into a "stronger" ensemble. Adaptive boosting is an approach to machine learning based on creating a highly accurate prediction rule by combining many relatively weak and inaccurate rules. Further on this, boosting involves creating a strong learner by iteratively adding weak learners and adjusting each weak learner's weight to focus on misclassified examples. It adapts to the hardness of each training sample. The AdaBoost algorithm of Schapire [98] was the first practical boosting algorithm and remained one of the most widely used and studied applications in numerous fields. Given a training set $(y_i, x_i)$, $i = 1, \ldots, n$, $x_i \in S$ and $y_i \in \{-1, 1\}$, for each learning round ($t = 1, \ldots, T$) using $m$ training examples, a distribution $D_i$ is computed (corresponding to the $P[Y|X]$) and a learning algorithm is applied to find a target function $h : S \to \{-1, 1\}$, where the aim of the weak learner is to find $h$ with low weighted error $\epsilon_i$ relative to $D_i$. The final result computes the sign of a weighted combination of weak classifiers:

$$f(x) = \sum_{t=1}^{n} \beta_i h_t(x_i). \tag{11}$$

Adaboost can be used to perform classification or regression. It can be understood as a procedure for greedily minimizing what has come to be called the exponential loss, namely:

$$\frac{1}{m} \sum_{i=1}^{m} exp(-y_i f(x_i)), \tag{12}$$

with $f$ introduced in the previous equation. In other words, it can be shown that the choices of $\beta_i$ and $h_i$ on each training round appear to be chosen so as to cause the most significant decrease in this loss.

In this paper, we use this approach to improve the classifier introduced in the random forest approach. In that case, the boosting method improves the convergence of the estimated regression function, using the new residuals of the proceeding leaf at each step. This being done many times. This algorithm uses an iterative process of convergence with residuals computed at each stage.

An introduction on this methodology is given by Ridgeway et al. [99]. Some recent prospects are described by Busa-Fekete et al. [100] and Ying et al. [101].

The AR(1) process is too simple for sophisticated data sets and cannot capture any nonlinear feature. Ridge regression allows us to improve the choice of the variables due to regularization. *k*-NN is interesting because its principle lies in using the variables whose properties are closer to the objective to attain. The algorithm for classification, like the random forest (can be used for discrete or continuous data) due to the splitting, permits scarcity and avoids overfitting. The boosting algorithm is iterative: at each step, it compares the re-estimation of the basic model with the previous error, and, if the base learner is easy to apply, the convergence to the objective is fast.

## 4. Data

To attain our objective, we need to seek the price discovery of Bitcoin. In this paper, we consider forecasts of both Bitcoin spot and futures systematically. Indeed, in a fundamental contribution, Baur and Dimpfl [102] indicates that the Bitcoin price discovery is led by the spot market and not by the futures markets, due to higher trading volume, 24/7 opening hours, and worldwide availability. Examining the interconnections between Bitcoin exchanges, Ji et al. [103] document that Coinbase is a strong leader in the market due to its popularity in the community, its US residence, and trading in US$. Our paper would contribute to the use of the adequate Bitcoin underlying for traders.

The Coinbase Bitcoin (CBBTCUSD) spot price in U.S. Dollars, in daily frequency, is pictured in Figure 2 from 13 January 2015 to 31 December 2020. The Bitcoin CME Futures contract of maturity December 2020 (BTZ20) is displayed since its creation in December 2017 on the Chicago Mercantile Exchange and the Chicago Board Options Exchange. During this period, six cryptocurrencies are available that we use to predict Bitcoin). We observe a specific economic cycle for Bitcoin: during the year 2018, we have a positive trend, and since the peak on December 20, 2018, we observe a decreasing trend. Thus, globally speaking this means that the Bitcoin prices evolve like a naive model such that if $Y_t$ represents the price at time $t$, then $Y_t = m_t + \varepsilon_t$, with $m_t = at + b$ and $\varepsilon_t$ a sequence of i.i.d. random variables. The trend characterized by the parameter *a* is positive during 2017 and negative during 2018. Then, a new economic cycle began in 2019, culminating in new "all-time highs" for Bitcoin by year's end of 2020. Notice, in this paper, we are not interested in modeling a possible bubble in 2018. We only focus on evolving the prices associated with an economic cycle (i.e., the classic phases of expansion, crisis, depression, and recovery). This exercise provides us a way to verify the accuracy of the forecasts done by different algorithms.

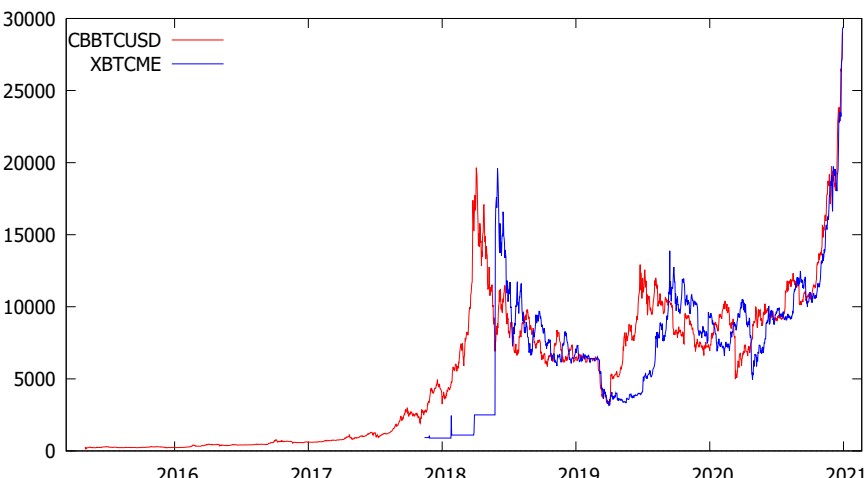

**Figure 2.** Coinbase Bitcoin spot price (red) and CME Bitcoin futures of maturity, December 2020 (blue) in US Dollars.

*4.1. Dataset Quality: A 'Financial Markets' Approach*

As opposed to a 'Blockchain' approach that would add, as an input to the machine learning models, non-standard econometric variables (e.g., hash rates, mining difficulty, block size, block version, number of transactions, the time between blocks, block size votes) about which we have neither theoretical grounding nor economic meaning virtually, we favor, in this paper, a 'Financial Markets' approach based on price relationships between various asset classes (e.g., stocks, bonds, foreign exchange, commodities, as well as other cryptocurrencies). Indeed, Koutmos [104] argues that Bitcoin prices, despite their seemingly attractive independent behavior relative to economic variables, may still be exposed to the same types of market risks which afflict the performance of conventional financial assets. According to Goldman Sachs, Bitcoin's 2021 returns even "destroy" everything on Wall Street, leading over assets from stocks to bonds, oil, banks, gold, and tech stocks (see Yahoo Finance (2021) at https://finance.yahoo.com/news/bitcoin-2021-returns-destroy-everything-223544895.html, accessed on 8 March 2021).

Table 1 details the daily data sourced from Coinbase and Datastream from 13 January 2015, to 31 December 2020. The 57 series cover cryptocurrencies, and traditional asset markets (stocks, bonds, foreign exchange), and commodities (e.g., energy, metals, grains, softs, cattle). The number of daily observations in this period is equal to 2070. Timestamps are converted to the European time zone to avoid look-ahead biases.

We also provide descriptive statistics for all the raw variables we consider in this exercise. They are listed in Table 2. They provide us some information regarding volatility, clustering, and extreme behavior. For instance, notice that the maximum Bitcoin spot price recorded is equal to 29,026$ by the year-end of 2020.

We introduce a cluster analysis to find groupings between all these variables as a classic unsupervised learning technique. In cluster analysis, data are partitioned into groups based on some measure of similarities or shared characteristics. Clusters are formed so that objects in the same cluster are very similar, and objects in different clusters are very distinct.

Louvain clustering detects communities in a network of nearest neighbors. More precisely, the Louvain clustering algorithm converts the dataset into a graph, where it finds highly interconnected nodes (Blondel et al. [105], Lambiotte et al. [106]). When applied to this dataset, this method confirms the existence of at least 11 different clusters.

The clustering of the dataset is represented graphically in Figure 3. This data projection by color regions reveals that cryptocurrencies (highlighted in yellow) tend to share common data attributes, especially Bitcoin with Monero, Tether, and Stellar. It also features several relatively segmented financial securities: international stocks, bonds, exchange rates, and commodities. This preliminary data analysis helps us detecting patterns depending on the asset class under consideration (e.g., stocks, bonds, commodities, or cryptocurrencies) that will be further assessed by the machine learning models.

**Table 1.** Database of crypto-assets, stocks, bonds, fiat-currencies and commodities.

| Asset Class | Name | Code |
|---|---|---|
| CRYPTO: | 1. Bitcoin Spot | CBBTCUSD |
| | 2. Bitcoin Futures | XBTCME |
| | 3. Ethereum | CBETHUSD |
| | 4. Ethereum Classic | ETHCLASSIC |
| | 5. Litecoin | CBLTCUSD |
| | 6. Bitcoin Cash | CBBCHUSD |
| | 7. Ripple | XRP |
| | 8. STELLAR | STELLAR |
| | 9. TETHER | TETHER |
| | 10. MONERO | MONERO |
| | 11. DASH | DASH |
| | 12. EOS | EOS |
| | 13. ZCASH | ZCASH |
| | 14. NEO | NEO |
| | 15. NANO | NANO |
| | 16. CARDANO | CARDANO |
| | 17. IOTA | IOTA |
| STOCKS: | 18. SP500 | STOCKSSP500 |
| | 19. VIX | VIXCLS |
| | 20. Dow Jones Industrial Average | DJIA |
| | 21. NASDAQ | NASDAQCOM |
| | 22. FTSE 100 | FTSE100 |
| | 23. Euro Stoxx 50 | STOXX50E |
| | 24. NIKKEI225 | NIKKEI225 |
| | 25. Shanghai Composite | SSEC |
| | 26. KOSPI | KS11 |
| | 27. Toronto Exchange TSX | GSPTSE |
| | 28. Bovespa Brazil | BVSP |
| BONDS: | 29. BAA Corporate Bond Yield relative to 10-Year Treasury rate | BONDSBAA10Y |
| | 30. 3-Month Treasury rate | DGS3MO |
| | 31. EURO BUND Futures | EUROBUND |
| | 32. 10-year Treasury Inflation-Indexed Security | DFII10 |
| FX: | 33. Trade-Weighted US Dollar Index | FXDTWEXM |
| | 34. US / Euro Foreign Exchange Rate | DEXUSEU |
| | 35. US / UK Foreign Exchange Rate | DEXUSUK |
| | 36. China / US Foreign Exchange Rate | DEXCHUS |
| ENERGY: | 37. Crude Oil WTI Futures | ENERGYDCOILWTICO |
| | 38. Ethanol Futures | ETHANOL |
| | 39. Gasoline Futures | RBF9 |
| | 40. Natural Gas Futures | NGAS |
| METALS: | 41. GOLD | METALSGOLDAMGBD228NLBM |
| | 42. SILVER | SILVER |
| | 43. ALUMINUM | ALUMINUM |
| | 44. COPPER | COPPER |
| | 45. LEAD | LEAD |
| | 46. NICKEL | NICKEL |
| | 47. ZINC | ZINC |
| GRAINS: | 48. US Corn Futures | GRAINSZCH9 |
| | 49. US Soybean Futures | ZSF9 |
| | 50. US Wheat Futures | ZWH9 |
| SOFTS: | 51. Orange Juice Futures | SOFTSOJF9 |
| | 52. US Cocoa Futures | CCH9 |
| | 53. US Coffee C Futures | KCH9 |
| | 54. US Cotton \#2 Futures | CTH9 |
| | 55. US Sugar \#11 Futures | SBH9 |
| CATTLE: | 56. Live Cattle Futures | CATTLELEG9 |
| | 57. Lean Hogs Futures | HEG9 |

**Table 2.** Descriptive statistics of the raw variables

| | Code | Mean | Median | Maximum | Minimum | Std. Dev. | Skewness | Kurtosis | Observations |
|---|---|---|---|---|---|---|---|---|---|
| 1 | ALUMINUM | 1821.02 | 1794.00 | 2539.50 | 175.25 | 200.24 | 0.04 | 5.00 | 1940 |
| 2 | BONDS_BAA10Y | 2.74 | 2.67 | 5.15 | 0.00 | 0.83 | −0.52 | 4.26 | 1940 |
| 3 | BVSP | 73,588.92 | 68,253.30 | 125,076.60 | 37,497.48 | 22,374.22 | 0.36 | 1.76 | 1940 |
| 4 | CARDANO | 0.13 | 0.08 | 1.10 | 0.02 | 0.14 | 3.62 | 19.10 | 990 |
| 5 | CATTLE_LEG9 | 488.12 | 295.11 | 3695.00 | 76.87 | 483.56 | 2.68 | 11.35 | 1115 |
| 6 | CBBCHUSD | 5240.90 | 5053.17 | 29,026.97 | 120.00 | 4949.29 | 0.94 | 4.06 | 2070 |
| 7 | CBETHUSD | 265.42 | 219.01 | 1386.02 | 0.00 | 235.91 | 1.43 | 5.48 | 1579 |
| 8 | CBLTCUSD | 65.14 | 52.26 | 359.40 | 0.00 | 55.18 | 1.81 | 7.42 | 1488 |
| 9 | CCH9 | 2589.58 | 2534.50 | 3410.00 | 1780.00 | 403.87 | 0.03 | 1.90 | 1940 |
| 10 | COPPER | 2.74 | 2.72 | 3.63 | 1.94 | 0.35 | −0.18 | 2.44 | 1940 |
| 11 | CBBTCUSD | 69.99 | 67.76 | 95.25 | 48.85 | 9.18 | 0.68 | 2.73 | 1940 |
| 12 | CTH9 | 207.36 | 118.34 | 1432.50 | 36.02 | 214.77 | 2.51 | 9.83 | 1292 |
| 13 | DASH | 55.66 | 53.19 | 107.95 | −36.98 | 20.17 | 0.43 | 4.52 | 1939 |
| 14 | DEXCHUS | 6.48 | 6.66 | 7.18 | 0.00 | 1.05 | −5.45 | 33.94 | 1940 |
| 15 | DEXUSEU | 1.13 | 1.13 | 1.39 | 0.00 | 0.19 | −4.52 | 27.88 | 1940 |
| 16 | DEXUSUK | 1.35 | 1.32 | 1.72 | 0.00 | 0.25 | −3.33 | 19.71 | 1940 |
| 17 | DFII10 | 0.55 | 0.45 | 2.82 | −1.08 | 0.84 | 0.85 | 3.97 | 1940 |
| 18 | DGS3MO | 0.84 | 0.34 | 2.47 | 0.00 | 0.86 | 0.56 | 1.72 | 1940 |
| 19 | DJIA | 21,566.14 | 21,730.87 | 31,097.97 | 0.00 | 5469.05 | −1.23 | 6.40 | 1940 |
| 20 | ENERGY_DCOILWTICO | 92.83 | 91.67 | 126.47 | 0.00 | 19.47 | −2.38 | 13.67 | 1940 |
| 21 | EOS | 4.66 | 3.58 | 21.42 | 0.49 | 3.32 | 1.69 | 6.27 | 1172 |
| 22 | ETHANOL | 1.52 | 1.47 | 3.52 | 0.82 | 0.28 | 1.91 | 9.62 | 1940 |
| 23 | ETHCLASSIC | 9.45 | 6.57 | 43.23 | 0.74 | 7.80 | 1.48 | 5.22 | 1511 |
| 24 | EUROBUND | 162.57 | 162.72 | 179.44 | 138.96 | 9.44 | −0.35 | 2.66 | 1940 |
| 25 | FTSE100 | 6887.99 | 6914.96 | 7877.45 | 4993.90 | 540.51 | −0.51 | 2.49 | 1940 |
| 26 | FX_DTWEXM | 1327.06 | 1289.38 | 2069.40 | 0.00 | 283.67 | −1.47 | 11.61 | 1940 |
| 27 | GRAINS_ZCH9 | 15,269.14 | 15,312.67 | 18,042.07 | 11,228.49 | 1189.11 | −0.28 | 2.68 | 1940 |
| 28 | GSPTSE | 72.36 | 67.74 | 133.38 | 37.33 | 17.30 | 1.33 | 5.01 | 1940 |
| 29 | HEG9 | 0.69 | 0.35 | 5.32 | 0.11 | 0.82 | 2.71 | 10.86 | 1190 |
| 30 | IOTA | 127.31 | 121.10 | 221.90 | 86.65 | 27.21 | 1.12 | 3.77 | 1940 |
| 31 | KCH9 | 2126.93 | 2063.05 | 3152.18 | 1457.64 | 204.33 | 1.25 | 5.38 | 1940 |
| 32 | KS11 | 2031.29 | 2030.88 | 2669.00 | 1563.50 | 234.66 | 0.36 | 2.57 | 1940 |
| 33 | LEAD | 123.56 | 119.71 | 171.00 | 83.83 | 18.65 | 0.63 | 2.59 | 1940 |
| 34 | METALS_GOLDAMGBD228NLBM | 67.14 | 49.60 | 475.00 | 0.10 | 78.35 | 1.84 | 6.94 | 2056 |
| 35 | MONERO | 2.35 | 1.10 | 20.46 | 0.35 | 3.23 | 3.04 | 12.63 | 963 |
| 36 | NANO | 6562.38 | 6365.63 | 13,201.98 | 0.00 | 2278.63 | 0.25 | 3.92 | 1940 |
| 37 | NASDAQCOM | 25.91 | 15.76 | 189.45 | 5.38 | 26.54 | 2.30 | 8.66 | 1104 |
| 38 | NEO | 2.85 | 2.76 | 6.15 | 1.48 | 0.74 | 1.03 | 4.27 | 1940 |
| 39 | NGAS | 13,039.22 | 12,930.00 | 21,174.00 | 7590.00 | 2774.20 | 0.31 | 2.46 | 1940 |
| 40 | NICKEL | 19,815.93 | 20,124.25 | 28,698.26 | 13,910.16 | 2929.49 | −0.11 | 2.41 | 1940 |
| 41 | NIKKEI225 | 135.73 | 136.00 | 232.85 | 91.25 | 27.22 | 0.61 | 3.29 | 1940 |
| 42 | RBF9 | 1.61 | 1.61 | 2.78 | 0.41 | 0.36 | 0.11 | 3.85 | 1940 |
| 43 | SBH9 | 14.32 | 13.73 | 23.81 | 9.21 | 2.79 | 0.96 | 3.53 | 1940 |
| 44 | SILVER | 17.16 | 16.66 | 29.26 | 11.81 | 2.64 | 2.14 | 8.14 | 1940 |
| 45 | SOFTS_OJF9 | 2463.59 | 2463.02 | 3824.68 | 0.00 | 618.50 | −1.13 | 6.92 | 1940 |
| 46 | SSEC | 2981.13 | 2999.48 | 5166.35 | 1991.25 | 506.87 | 0.45 | 4.99 | 1940 |
| 47 | STELLAR | 0.13 | 0.08 | 0.89 | 0.00 | 0.12 | 1.84 | 7.18 | 1302 |
| 48 | STOCKS_SP500 | 3320.98 | 3333.71 | 3865.18 | 2385.82 | 247.03 | −0.28 | 2.55 | 1940 |
| 49 | STOXX50E | 1.00 | 1.00 | 1.06 | 0.90 | 0.01 | −2.54 | 20.59 | 1250 |
| 50 | TETHER | 16.59 | 14.47 | 82.69 | 0.00 | 7.96 | 2.69 | 16.77 | 1940 |
| 51 | VIXCLS | 8111.91 | 8100.00 | 29,385.00 | 888.00 | 4311.86 | 1.08 | 6.05 | 1143 |
| 52 | XBTCME | 0.24 | 0.20 | 2.78 | 0.00 | 0.31 | 3.17 | 19.44 | 2064 |
| 53 | XRP | 143.53 | 74.55 | 1900.00 | 24.56 | 146.44 | 3.66 | 30.31 | 1418 |
| 54 | ZCASH | 374.88 | 368.75 | 515.75 | 301.50 | 35.86 | 1.30 | 5.41 | 1940 |
| 55 | ZINC | 2383.54 | 2327.50 | 3579.50 | 1460.50 | 430.92 | 0.43 | 2.88 | 1940 |
| 56 | ZSF9 | 942.31 | 920.62 | 1310.25 | 803.50 | 80.27 | 1.14 | 4.58 | 1580 |
| 57 | ZWH9 | 505.81 | 504.44 | 738.88 | 384.12 | 63.07 | 0.74 | 3.82 | 1940 |

Source: Coinbase and Datastream.

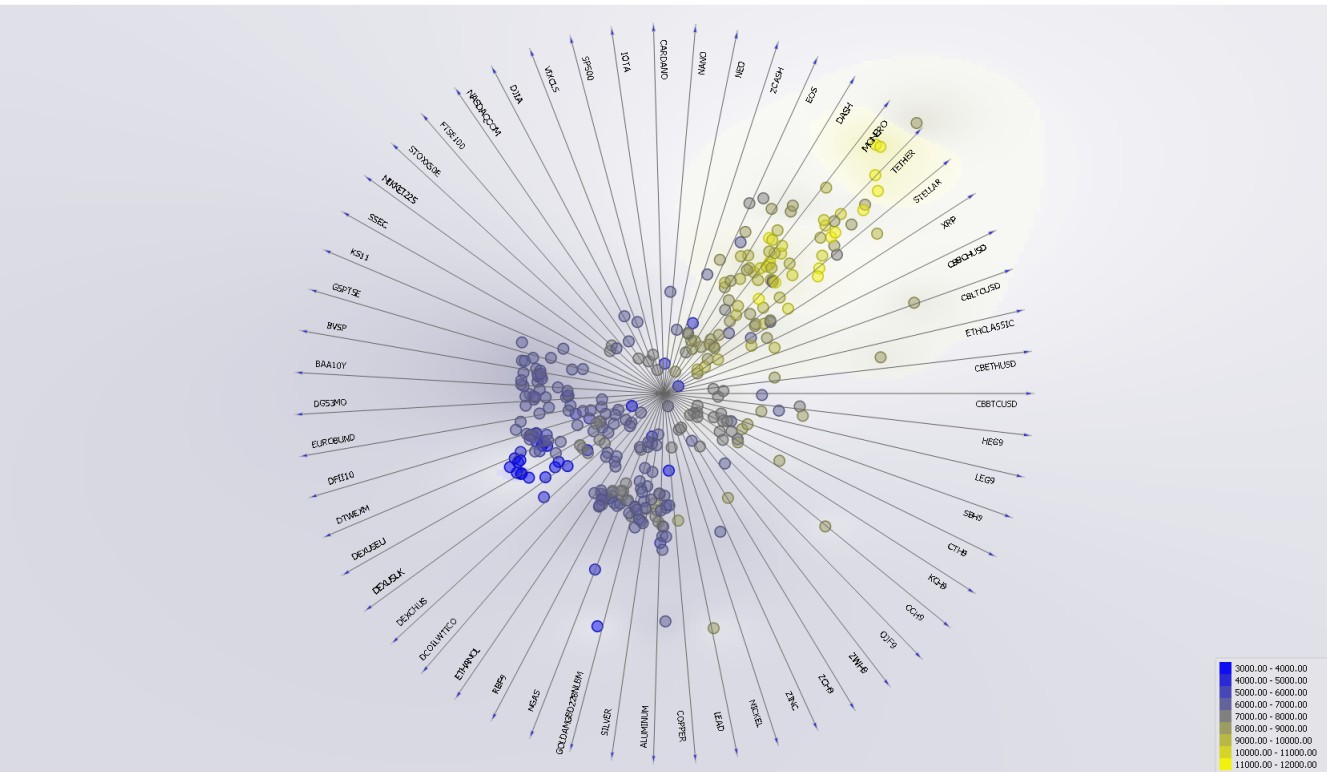

**Figure 3.** Louvain Clustering. Note: In Louvain clustering, principal components analysis processing is typically applied to the original data to remove noise. Following modularity optimization, the detection algorithm then unfolds to retrieve clusters and produce graphs of highly interconnected nodes. Modularity is a scale value between −1 and 1 that measures the density of edges inside communities to edges outside communities. Optimizing this value theoretically results in the best possible grouping of the nodes of a given network. However, going through all possible iterations of the nodes into groups is impractical. Therefore, heuristic algorithms are used (see, e.g., Clauset et al. [107] for more details).

### 4.2. Sub-Samples Decomposition

We specify the different periods on which we analyze all the datasets.

- Different sets of data: To predict the Bitcoin price, described as a cryptocurrency without clearly established fundamentals, we follow an approach based on financial markets. That is to say, we include in the pool of predictors several underlying and analyze their contribution in explaining Bitcoin. As shown in Table 1, we retain 56 variables (besides the Bitcoin price) that belong to the following categories: (i) cryptocurrencies, (ii) stocks, (iii) bonds, (iv) foreign exchange rates, and (v) commodities. To predict Bitcoin following the schemes proposed in Section 3, $\hat{Y}_t = f(X_t)$, we proceed step-by-step for the choice of the variables using up to five different vectors $X_t$: (1) $X_t$ composed of sixteen cryptocurrencies, (2) $X_t$ composed of eleven stocks, (3) $X_t$ composed of four bonds, (4) $X_t$ composed of four foreign exchange rates, (5) $X_t$ composed of twenty-one commodities, and (6) $X_t$ is composed of fifty-six variables. For each step, we train and test the samples using seven modelings (one AR(1) and six machine learning algorithms, see Section 5.1 for details). This approach permits us to detect each subsample of variables $X_t$ in the forecasts of Bitcoin.
- Training and testing set: for each period considered, we need to specify the length of the training set (e.g., a known set of input data) and the testing set (e.g., new input data) to test the models' predictions.
- The choice of the period:
    1. We consider the whole sample from 13 January 2015 to 31 December 2020. In this sample, we use only six cryptocurrencies (Litecoin, Ethereum, Stellar, Ripple,

Monero, Dash). We train the inputs $X_t$ from 13 January 2015 to 31 December 2016 and then test our predictions from 1 January 2017 to 31 December 2020.

2.   As robustness checks, we further assess the accuracy of our predictions on four sub-samples.

   (a)   We favor the availability of cryptocurrency prices during 24 January 2018 to 31 December 2020 to include up to 17 (some newly created) cryptocurrencies (Bitcoin Spot, Bitcoin Futures, Ethereum, Ethereum Classic, Litecoin, Bitcoin Cash, Ripple, Stellar, Tether, Monero, Dash, EOS, Zcash, Neo, NANO, Cardano, IOTA). $T_{train}$ = 24 January 2018 to 31 December 2018. $T_{test}$ = 1 January 2019 to 31 December 2020.

   (b)   We introduce the stable coin (any crypto-currency pegged to either fiat currency or government-backed security (like a bond) counts as a stable coin. The idea is that this crypto-currency will be more stable or less volatile. Asset-backed cryptocurrencies are not necessarily centralized since there may be a decentralized vaults and commodity holders network rather than a centralized controlling body. The advantages of asset-backed cryptocurrencies are that coins are stabilized by assets that fluctuate outside the cryptocurrency space reducing financial risk. The Tether currency is backed by the dollar (1:1). For more details, we refer to Abraham and Guegan [108]). Tether (rumored 1 US\$ = 1 Tether) available since 12 April 2017. $T_{train}$ = 12 April 2017 to 30 November 2018. $T_{test}$= 1 December 2018 to 31 December 2020.

   (c)   We consider a 'classical economic cycle' (e.g., expansion-crisis-depression-recovery) for Bitcoin during the years 2016 to 2018. $T_{train}$ = 01 January 2016 to 31 December 2016. $T_{test}$= 01 January 2017 to 31 December 2018.

   (d)   Lastly, we use the last historical year of trading to make predictions. $T_{train}$ = 1 January 2019 to 30 June 2019. $T_{test}$= 01 July 2019 to 31 December 2020.

### 4.3. Software

We detail the software used to perform this exercise. Pre-processing of the data and ML algorithms are entirely conducted in Python 3.6 (or newer) with Anaconda Navigator, relying on the following libraries:

- `Timeseries.ARIMA` (AR(1)),
- `classification.neuralnetwork.MLPClassifierWCallback/NNClassificationLearner` (Artificial Neural Network with Multi-Layer Perceptron),
- `sklearn.ensemble.forest.RandomForestClassifier/Learner` (Random forest),
- `sklearn.svm.classes.SVC` (Support Vector Machines) based on `libsvm`,
- `sklearn.neighbors.classification.KNeighborsClassifier/KNNLearner` (K-Nearest Neighbors),
- `SAMME.R` (AdaBoosting method),
- `regression.linear.LinearRegressionLearner/ridgelambda` with lambda the parameter controlling the regularization (Ridge regression).

Several other functions are used (such as `functions.rmse(true,pred)` (Root mean squared error), `functions.mape(true,pred)` (Mean absolute percentage error) `functions.mae (true,pred)` (Median absolute error)), but they are not displayed here for space constraints.

## 5. Main Results

This section contains the Bitcoin price predictions (spot and futures) based on machine learning techniques, i.e., the forecasts occurring during the testing period for each of the seven algorithms. For each algorithm, we specify the parameters we used.

*5.1. Parameterization*

Estimation accuracy depends on an appropriate set of parameters. When the choices of parameters are not precisely documented, that is, precisely where the 'risks' of machine learning arise according to Abadie and Kasy [11]. We predict the Bitcoin price (spot and futures) based on seven competing algorithms for which we provide the parameterization details thoroughly:

1. AR(1): the autoregressive regression of order one is estimated and tested to predict the Bitcoin spot or futures price.
2. Artificial Neural Network: to predict Bitcoin, we choose the perceptron algorithm with backpropagation. We compute 200 iterations with 10 neurons in the hidden layer. The ReLu activation is used. The optimizer is the Adam solver, and the regularization parameter is set to 0.0001.
3. Random Forest: we predict Bitcoin using an ensemble of 10 decision trees, with a depth of 3 trees. The stopping parameter is $10^{-3}$.
4. SVM: the support vector machine inputs to higher-dimensional feature spaces. To predict Bitcoin, we resort to the RBF kernel, with 100 iterations, the cost set to $c = 1$, and the parameter $\epsilon$ set to 0.1. The cost is a penalty term for loss and applies to classification and regression tasks. In SVM, $\epsilon$ applies to the regression tasks. It defines the distance from true values within which no penalty is associated with predicted values.
5. kNN: we predict Bitcoin according to the nearest training distances using three neighbors with uniform weight, as measured by the Euclidean metric. Figure 4 contains the Density-based spatial clustering from which the number of neighbors has been detected.
6. AdaBoost: this ensemble meta-algorithm combines weak learners and adapts to the 'hardness' of each training sample. The boosting is performed thanks to the SAMME.R classification algorithm, which exhibits a linear regression loss function.
7. Ridge regression: this latter method minimizes an objective function using a stochastic approximation of gradient descent. In the classification (Hinge) and regression (Squared) loss functions, $\epsilon$ is set to 0.10. The Ridge L2 regularization is used (Lasso and elastic net) with strength 0.00001, mixing 0.15, constant learning rate, 0.01 initial learning rate, 1000 iterations, and a stopping criterion set at 0.001. To predict Bitcoin, we shuffle data after each iteration.

Forecasting is based on 10-fold stratified cross-validation, with the training set size is set at 66%, and the repeated sequence between training and test samples is set at 10.

Forecast Statistics

To discriminate between competing forecasts, we compare the accuracy of the predictions using the Root Mean Square Error (RMSE), the Mean Absolute Error (MAE), and the Mean Absolute Percent Error (MAPE). Suppose the forecast sample is $j = T + 1, T + 2, \dots, T + h, h \in N$, and denote the actual and forecast value in period $t$ as $Y_t$ and $\hat{Y}_t$, respectively:

$$RMSE = \sqrt{\frac{\sum_{t=T+1}^{T+h} (\hat{Y}_t - Y_t)^2}{h}} \tag{13}$$

$$MAE = \sum_{t=T+1}^{T+h} \frac{|\hat{Y}_t - Y_t|}{h} \tag{14}$$

$$MAPE = 100 \times \sum_{t=T+1}^{T+h} \frac{\left|\frac{\hat{Y}_t - Y_t}{y_t}\right|}{h}. \tag{15}$$

The best forecasts are obtained by minimizing these forecast evaluation statistics.

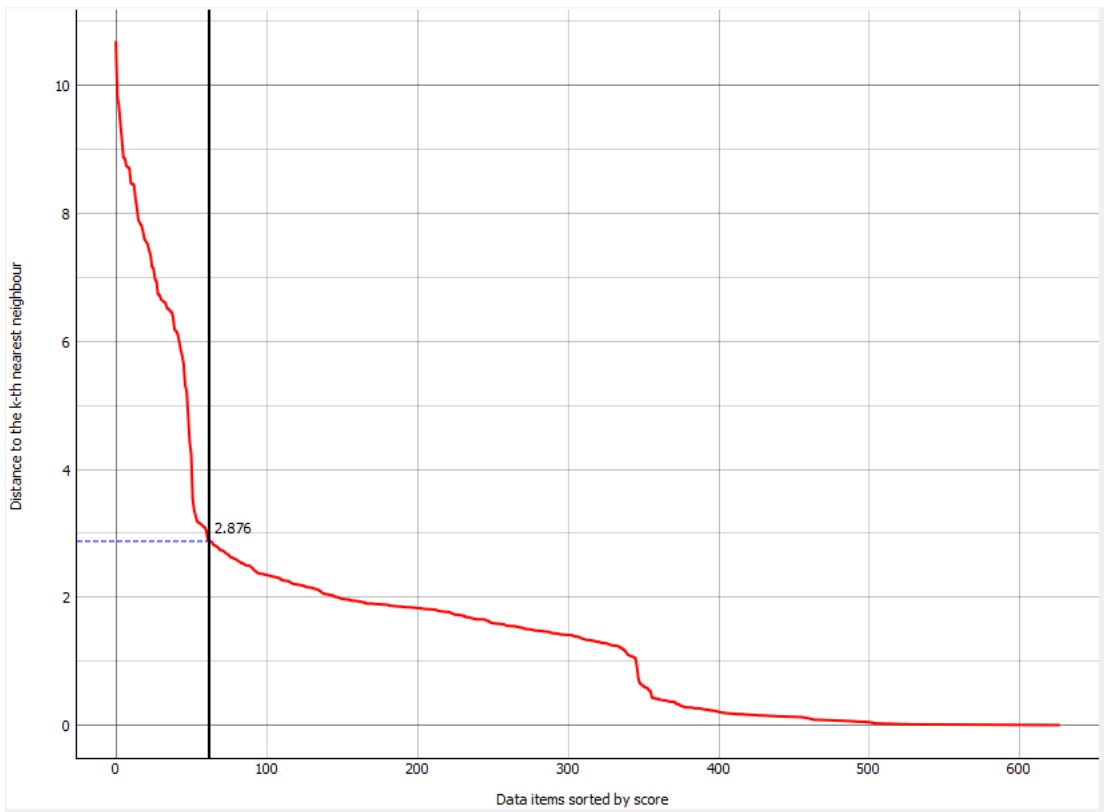

**Figure 4.** DBSCAN clustering algorithm for Bitcoin k-NN distance setting. Note: The density-based spatial clustering algorithm gives the idea of an ideal selection for the neighborhood distance setting. Here, 2.876 has been rounded up to 3 in the empirical application.

### 5.2. Forecasting Results for the Coinbase Bitcoin Spot Price

To discriminate between the variables which increase the accuracy of the predictions and those which pollute them, we consider the different sets of forecasting results provided in Table 3. (1) In column AR(1), we provide the predictions obtained using only the past of Bitcoin. (2) The first row, 'crypto', provides the results using six cryptocurrencies (Litecoin, Ethereum, Stellar, Ripple, Monero, Dash). (3) The second row, 'stocks', provides the results using the traditional financial assets. (4) The third row, 'commo', provides the results obtained with the commodities. (5) The fourth row, 'all', provides the results using the whole data set. We always give the results for the three previous criteria.

Main Results

In Table 3, we provide in each column the errors computed by Equations (13)–(15). The smallest result of these three criteria provides the best forecast for different approaches and a given set of variables. The best spot price predictions are achieved, respectively, by the algorithms Adaboost, Random forest, and kNN. The MAPE is inferior both to RMSE and MAE. Apart from this group, SVM and ANN provide worse forecasting results. The AR(1) suggests that only considering the past of Bitcoin historical prices is not relevant.

If we compare the forecasts across categories, we conclude that using the information embedded in cryptocurrencies is enough to predict Bitcoin. Indeed, adding other financial securities does not improve the forecasting error. For instance, the MAPE for cryptocurrencies is equal to 0.15, which is inferior to that of Stocks (0.36) and Commodities (0.39).

**Table 3.** Full sample forecasting results for the Coinbase Bitcoin spot price.

| CRYPTO → SPOT BTC | AR(1) | ann | random | svm | knn | boost | ridge |
|---|---|---|---|---|---|---|---|
| **RMSE** | 1207.51 | 1607.31 | 137.65 | 1043.32 | 395.29 | 23.42 | 304.77 |
| **MAE** | 990.45 | 1185.68 | 88.54 | 772.18 | 280.84 | 9.94 | 238.1 |
| **MAPE** | 14.29 | 19.97 | 1.22 | 12.96 | 3.45 | 0.15 | 3.54 |
| **STOCKS-BONDS-FX** | AR(1) | ann | random | svm | knn | boost | ridge |
| **RMSE** | 1207.51 | 1652.17 | 221.95 | 1065.93 | 589.97 | 46.21 | 915.98 |
| **MAE** | 990.45 | 1215.37 | 139.48 | 829.05 | 399.96 | 21.39 | 751.81 |
| **MAPE** | 14.29 | 19.52 | 1.93 | 12.59 | 6.082 | 0.36 | 10.58 |
| **COMMO** | AR(1) | ann | random | svm | knn | boost | ridge |
| **RMSE** | 1207.51 | 1112.14 | 207.4 | 848.86 | 532.17 | 54.86 | 793.8 |
| **MAE** | 990.45 | 817.07 | 141.96 | 684.43 | 352.37 | 24.94 | 639.16 |
| **MAPE** | 14.29 | 12.95 | 1.96 | 10.36 | 4.61 | 0.39 | 8.86 |
| **ALL** | AR(1) | ann | random | svm | knn | boost | ridge |
| **RMSE** | 1207.51 | 1484.09 | 131.27 | 938.52 | 463.32 | 19.18 | 230.12 |
| **MAE** | 990.45 | 1117.8 | 85.32 | 704.11 | 285.23 | 7.23 | 180.67 |
| **MAPE** | 14.29 | 18.84 | 1.13 | 11.65 | 4.08 | 0.11 | 2.6 |

Note: AR(1) stands for the autoregressive model of order one; ann for the Artificial Neural Network model; random for the Random forest model; svm for the Support Vector Machine model; knn for the k-Nearest neighbor model; boost for the Adaboost model; and ridge for the Ridge regression. In terms of forecast statistics, we resort to the Root Mean Square Error (RMSE), the Mean Absolute Error (MAE), and the Mean Absolute Percent Error (MAPE).

From this first round of results dedicated to Bitcoin spot prices, we obtain a similar conclusion as Klein et al. [109]: "*Bitcoin as an asset does not resemble any other conventional asset from an econometric perspective.*" Thanks to this result, we begin to specify the characteristics of this asset.

Next, if we use the whole database of 56 series, we acknowledge that the smallest forecast errors are reached in this setting: MAPE all = 0.11 < MAPE cryptocurrencies = 0.15; RMSE all = 19.18 < RMSE cryptocurrencies = 23.42. Thus, we note that over this period, using an extensive set of information improves the forecast of the Bitcoin, with the Adaboost algorithm, in the sense of the forecast error. The regression with a Ridge regularization giving a very indirect result.

When identifying the relevant variables behind the Bitcoin spot price variations, it appears that the MAPE of cryptocurrencies only is, therefore, very satisfactory (0.15). Adding many of the 50 other financial series improves the forecast error marginally (e.g., MAPE = 0.11).

### 5.3. Forecasting Results for the CME Bitcoin Futures Price

Next, we investigate the forecasts of the Bitcoin futures contract of maturity December 2019 in Table 4. The results are close to spot forecasting since the algorithm providing the smallest prediction errors is still Adaboost, then Random forest. In that run, the AR(1) ends in third position (surprisingly for predicting the future) when inspecting, for instance, the MAPE. Nevertheless, when comparing Tables 3 and 4, we observe that the forecasting errors are much larger for futures using all algorithms (except AR(1)).

Looking at the two Tables 3 and 4, we observe that: (i) It seems to exist a clear segmentation between Cryptocurrencies, Financial and Alternative assets. Forecasting is only slightly improved by adding step-by-step further variables. (ii) Contrary to literature (Kapar and Olmo [110], Entrop et al. [111]), we do not identify clear price fundamentals for Bitcoin. (iii) Bitcoin does not seem to be integrated into commodities. (iv) Bitcoin does not seem integrated into financial asset markets. (v) Our paper underlines the need for the researcher to implement sparse models and not falling into the trap of overfitting (Athey and Imbens [12] already discuss the sparsity in machine learning versus econometric models).

**Table 4.** Full sample forecasting results for the CME Bitcoin December 2019 Futures price.

| CRYPTO → FUT BTC | AR(1) | ann | random | svm | knn | boost | ridge |
|---|---|---|---|---|---|---|---|
| RMSE | 1050.27 | 3186.26 | 546.17 | 2943.21 | 1616.74 | 122.45 | 1568.51 |
| MAE | 402.17 | 2300.306 | 272.66 | 2093.4 | 771.56 | 64.88 | 1147.17 |
| MAPE | 5.72 | 35.98 | 4.15 | 35.23 | 12.53 | 0.99 | 17.54 |
| STOCKS-BONDS-FX | AR(1) | ann | random | svm | knn | boost | ridge |
| RMSE | 1050.27 | 2742.52 | 558.81 | 2619.05 | 1102.24 | 88.31 | 2228.131 |
| MAE | 402.17 | 1921.73 | 304.33 | 1807.49 | 566.96 | 37.41 | 1576.706 |
| MAPE | 5.72 | 28.1 | 4.7 | 26.94 | 8.36 | 0.61 | 25.59 |
| COMMO | AR(1) | ann | random | svm | knn | boost | ridge |
| RMSE | 1050.27 | 2598.26 | 590.71 | 2693.82 | 1403.53 | 79.09 | 2341.55 |
| MAE | 402.17 | 1784.42 | 301.08 | 1877.07 | 722.64 | 40.45 | 1660.21 |
| MAPE | 5.72 | 29.61 | 4.26 | 32.11 | 11.79 | 0.7 | 28.33 |
| ALL | AR(1) | ann | random | svm | knn | boost | ridge |
| RMSE | 1050.27 | 2795.35 | 369.63 | 2625.96 | 938.52 | 86.15 | 1317.23 |
| MAE | 402.17 | 1984.95 | 187.86 | 1850.06 | 426.76 | 44.53 | 915.82 |
| MAPE | 5.72 | 34.6 | 2.62 | 32.62 | 6.075 | 0.73 | 15.56 |

Note: AR(1) stands for the autoregressive model of order one; ann for the Artificial Neural Network model; random for the Random forest model; svm for the Support Vector Machine model; knn for the k-Nearest neighbor model; boost for the Adaboost model; and ridge for the Ridge regression. In terms of forecast statistics, we resort to the Root Mean Square Error (RMSE), the Mean Absolute Error (MAE), and the Mean Absolute Percent Error (MAPE).

## 5.4. Visualization

In this section, we resort to unsupervised learning to find hidden patterns in the input data. As advocated by Zhao and Hastie [8], visualization allows us to check whether there are differences in interpretation between several kinds of data inspection tools and the actual results from the machine learning models.

We confirm these results by inspecting Sieve diagrams (Riedwyl and Schüpbach [112])—which allow visualizing the observed and expected frequencies between pairs—in Figure 5. Bitcoin and Litecoin (which are based on the same protocol) display similar characteristics (highlighted in dark blue and red colors), whereas Bitcoin is found merely different from other assets (say S&P 500, US 10-Year rate, US Dollar or Oil price) as judged by the light (blue and red) colors.

Figure 6 displays a self-organizing map (Kohonen [113]), i.e., a neural-network-based clustering that transforms a dataset into a topology-preserving two-dimensional map. We use a neighborhood function to preserve the topological properties of the input space. When applied to the Bitcoin spot, it confirms its shared characteristics with most of the other cryptocurrencies (in dark yellow, arranged at the beginning of the database, according to Table 1). On the contrary, virtually no connection is visible with traditional asset markets (in grey at the database center). The map picks up some interest between Bitcoin and commodities at the end of the database (in light yellow color).

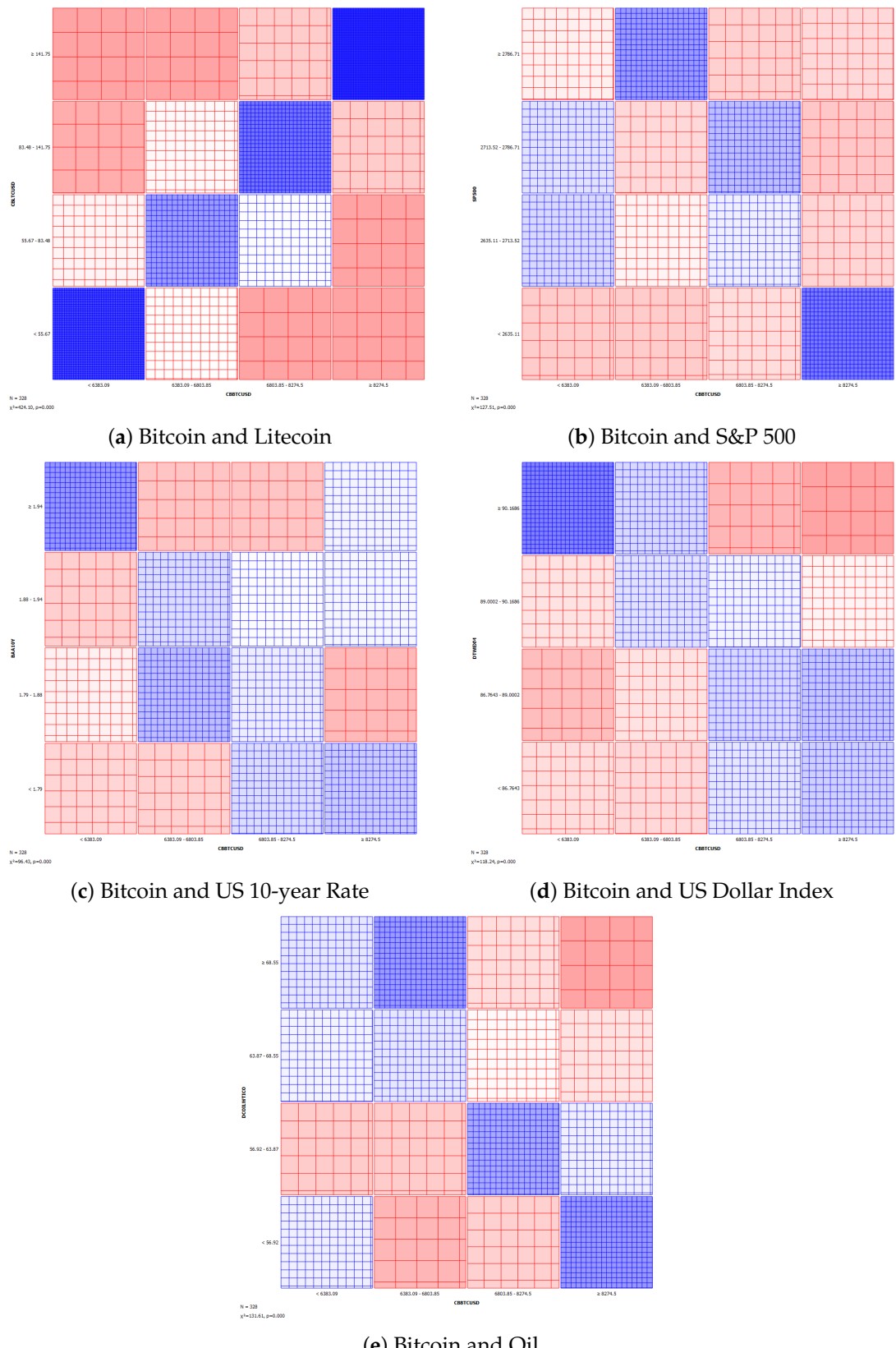

(**a**) Bitcoin and Litecoin

(**b**) Bitcoin and S&P 500

(**c**) Bitcoin and US 10-year Rate

(**d**) Bitcoin and US Dollar Index

(**e**) Bitcoin and Oil

**Figure 5.** Sieve diagrams between Bitcoin spot and other assets. Note: In Sieve or 'parquet' diagrams, the area of each rectangle is proportional to the expected frequency, while the observed frequency is shown by the number of squares in each rectangle. The difference between observed and expected frequency (proportional to the standard Pearson residual) appears as the density of shading, using color to indicate whether the deviation from independence is positive (blue) or negative (red).

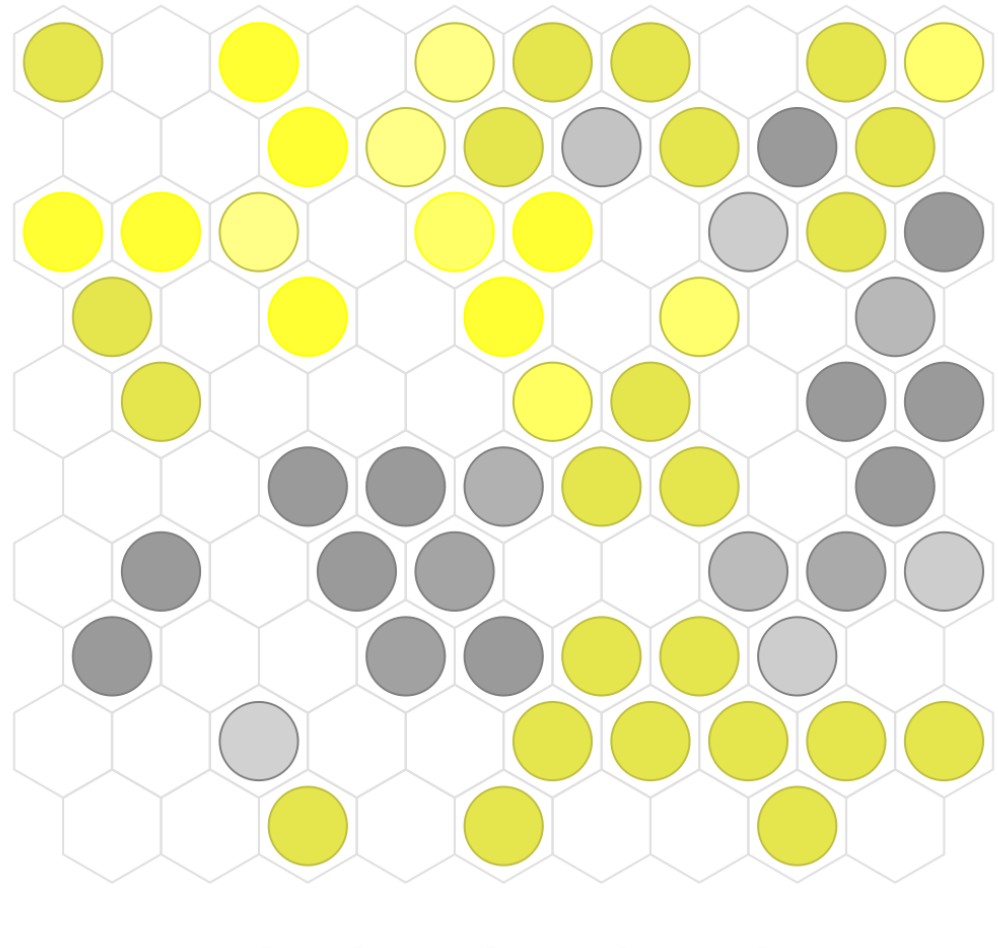

**Figure 6.** Self-organizing Map of Bitcoin spot with the rest of the database. Note: A self-organizing map is an unsupervised learning algorithm that infers low, typically a two-dimensional discretized representation of the input space called a map. The map preserves the topological properties of the input space. The data are ranked according to Table 1. In yellow color, the variables share similar characteristics with the Bitcoin spot (mostly cryptos). In grey color, the variables appear to share lower characteristics with Bitcoin. Against the white background, the variables appear to exhibit no relationship with Bitcoin.

The same information is mainly conveyed by the Multi-Dimensional Scaling map (Wickelmaier [114]) pictured in Figure 7, where each series' position at each time interval corresponds to the sum of forces acting on it (pushing the series apart or together concerning Bitcoin). The center of the map captures strong interrelations between cryptocurrencies (in yellow). The edges of the map (in blue) delimit other areas of strong interrelations between traditional assets (crossed dots on the left-hand side) and commodities (blue stars on the right-hand side). Notice that commodities are somewhat located closer to Bitcoin than traditional assets in that latter example.

As robustness check, a *t*-distributed Stochastic Neighbor Embedding (Maaten and Hinton [115], Van Der Maaten [116]) map reveals the same kind of information in Figure 8: cryptocurrencies (in yellow) are represented in the center of the map with respect to the high interconnection level with Bitcoin, whereas other assets (financial and commodities) form two other distinct regions (in blue) on the edges.

Taken together, these various visualization plots have reinforced the impression of segmentation of Bitcoin and cryptocurrencies altogether versus traditional financial assets and commodities.

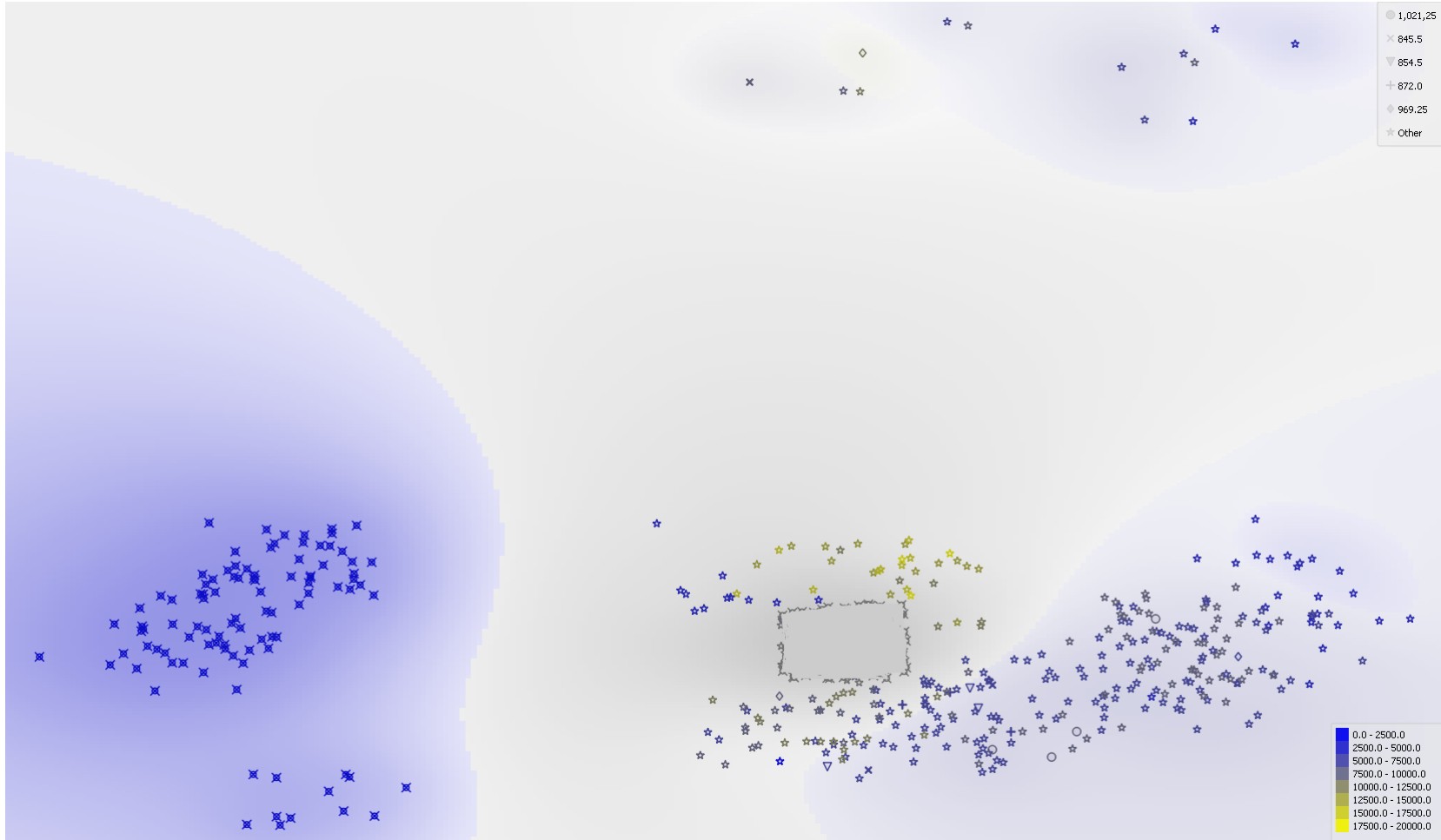

**Figure 7.** Multi-Dimensional Scaling map concerning Bitcoin spot. Note: multi-dimensional scaling is a technique that finds a low-dimensional (e.g., two-) projection of points, where it tries to fit distances between points as well as possible. The algorithm iteratively moves the points around in a kind of a simulation of a physical model. If two points are too close to each other (or too far away), there is a force pushing them apart (or together). The change of the point's position at each time interval corresponds to the sum of forces acting on it.

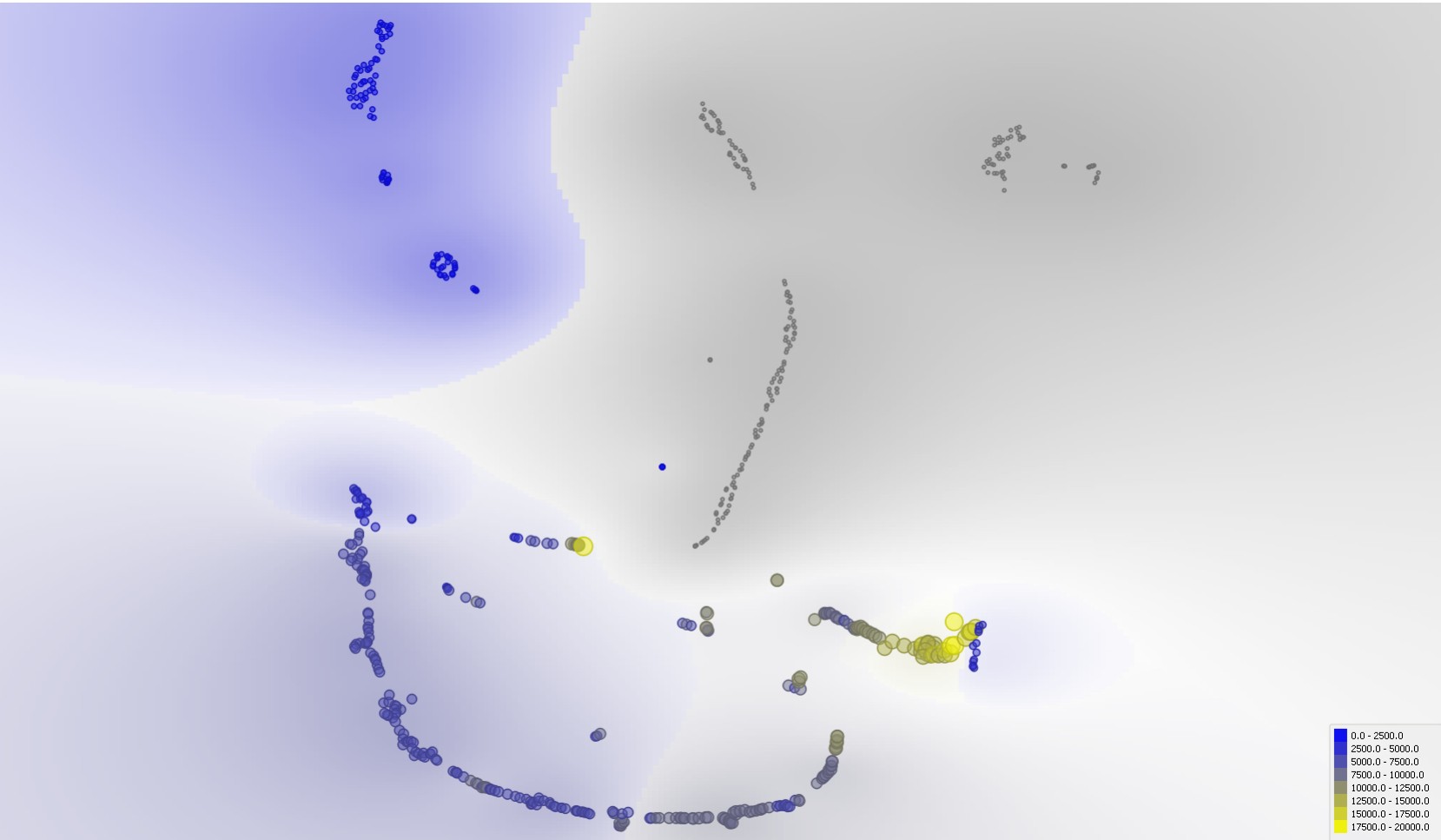

**Figure 8.** *t*-distributed stochastic neighbor-embedding projection with respect to the Bitcoin spot price. Note: *t*-distributed stochastic neighbor embedding is a dimensionality reduction technique, similar to multi-dimensional scaling, where points are mapped to 2D space by their probability distribution.

## 6. Robustness Checks

Full sample results convey the idea that using cryptocurrencies only is enough to predict Bitcoin. Adding other variables from the financial markets deteriorates the forecasting accuracy.

In Section 5, we have established that the Bitcoin spot's predictive strategies and futures are not the same. On the one hand, for the Bitcoin spot during the whole sample (13 January 2015 to 31 December 2020), we have identified that cryptocurrencies are segmented from traditional financial and commodity markets. On the other hand, from the creation of Bitcoin futures in December 2017 until the end of our database in December 2020, it appears that CME's Bitcoin derivatives instrument is better explained by stocks and commodities. Maybe because the CME also heavily trades futures for stocks and commodities (see the discussion on the birth of the Bitcoin futures market in Baur and Dimpfl [102]).

In what follows, we introduce several subsample forecasts for sensitivity purposes of our main previous results. In Section 6.1, we forecast Bitcoin spot and futures by using only cryptocurrencies (newest to date) starting on 24 January 2018. In Section 6.2, we study the influence of Tether on the market as a means to print US$ and convert them to cryptocurrencies. Section 6.3 follows a classic business cycle (expansion/contraction) in the Bitcoin price in the year 2016–2018. In Sections 6.4 and 6.5, we harness our results against the last two years of historical prices available for trading. Section 6.6 develops trading strategies.

### 6.1. 'Crypto Select'

Since the overarching result of our paper is that the price of Bitcoin appears somewhat disconnected from commodities and traditional asset markets (in terms of additional forecasting power), we resort to subsample estimates across the maximum of data available for other cryptocurrencies. The retained sample is 24 January 2018 to 11 December 2019 to include the six newest cryptocurrencies (effectively collecting the seventeen cryptocurrencies listed in Table 1 for this sensitivity analysis).

Regarding Table 5, we confirm the excellent performance of AdaBoost in terms of most minor errors, followed closely by Random forest and K-nearest neighbors, even if we change the underlying set of cryptocurrencies. Concerning the futures contract, the forecast errors are negligible when we use seventeen cryptocurrencies instead of six (0.51 < 0.99). The forecast errors for the Bitcoin spot are smaller when we use only six cryptocurrencies rather than seventeen (0.24 > 0.15). However, these spreads are not large, and they are inferior for a spot than for futures (which have been created recently and for which we have fewer information). From a trading perspective, it seems that it could be interesting to use the maximum of variables (to have various pairs of cryptocurrencies to trade).

**Table 5.** Bitcoin Coinbase Spot and CME Futures Forecasts based on the most recent 17 cryptocurrencies only from 24 January, 2018, to 11 December, 2019.

| CRYPTO → SPOT BTC | AR(1) | ann | random | svm | knn | boost | ridge |
|---|---|---|---|---|---|---|---|
| RMSE | 2099.45 | 1882.68 | 215.45 | 1909.24 | 364.80 | 39.24 | 1249.79 |
| MAE | 1714.52 | 1399.44 | 131.41 | 1548.98 | 231.11 | 16.04 | 955.87 |
| MAPE | 26.61 | 21.34 | 1.64 | 22.34 | 2.86 | 0.24 | 13.20 |
| CRYPTO → FUT BTC | AR(1) | ann | random | svm | knn | boost | ridge |
| RMSE | 2880.50 | 2759.37 | 417.65 | 2685.02 | 825.13 | 76.98 | 1835.26 |
| MAE | 2146.53 | 2065.76 | 239.82 | 2002.00 | 415.26 | 32.37 | 1389.52 |
| MAPE | 32.29 | 33.84 | 3.25 | 32.88 | 6.65 | 0.51 | 20.02 |

Note: AR(1) stands for the autoregressive model of order one; ann for the Artificial Neural Network model; random for the Random forest model; svm for the Support Vector Machine model; knn for the k-Nearest neighbor model; boost for the Adaboost model; and ridge for the Ridge regression. In terms of forecast statistics, we resort to the Root Mean Square Error (RMSE), the Mean Absolute Error (MAE), and the Mean Absolute Percent Error (MAPE).

*6.2. 2017 Tether's Introduction*

We now consider a smaller period of training to predict the Bitcoin prices to introduce inside the set of variables the stable coin Tether (1 USDT = 1 US$). Indeed, this cryptocurrency is valuable on derivatives exchanges such as Bitrex and Bitfinex, when the investors exchange their cryptocurrency into fiat currency.

In a seminal paper on BitMEX bitcoin derivatives (off-shore, unregulated) exchanges, Alexander et al. [117] recalls the speculative role played by Tether in the trading community as a means of exchanging 1 US$ against 1 USD Tether (USDT, supposedly backed one-to-one in BitMEX). Chief concerns among market players is the possibility of "front-running" by printing large amounts of USD into Bitcoin to provoke a price jump, and then once the market is shocked, revert back the transaction from Bitcoin to USD to effectively cash in your benefit. This illegal procedure through Tether is documented by Griffin and Shams [118]. Besides, Tether is supposed to maintain a 1-to-1 ratio to be pegged against the USD. Bitfinex is accused of unlawfully covering up the true levels of its currency reserves. The exchange, owner of Tether, was prosecuted in the USA. On 23 February 2021, the New York attorney general's office settled a nearly two-year investigation into the finances and corporate practices of the companies that operate the Bitfinex cryptocurrency exchange and the stablecoin Tether The Hong Kong-based iFinex Inc., which operates the Bitfinex exchange, and Tether Ltd. agreed to pay $18.5 million to the attorney general's office. See the Wall Street Journal (2021) at https://www.wsj.com/articles/cryptocurrency-firms-bitfinex-tether-settle-new-york-attorney-generals-probe-11614093709, accessed on 23 February 2021).

Tether has begun to be exchanged on 12 April 2017. Thus, we constitute a subsample from 12 April 2017 to 31 December 2020. Compared to full-sample results, this subperiod allows us to increase the set of cryptocurrencies, introducing the cryptocurrencies Dash and Stellar. Then, we perform the same analysis as in Section 5 using the machine learning models on 12 April 2017–30 November 2018 as a training period and until the end of December 2020 to test the models' accuracy. As before, we introduce the input variables step by step : (1) the Bitcoin (spot and futures), (2) nine available cryptocurrencies (i.e., Litecoin, Ethereum, Ethereum Classic, Ripple, Stellar, Tether, Monero, Dash, Zcash), (3) all the traditional financial assets, (4) all the commodities, (5) all the sets.

Sub-Sample Results

Looking at Tables 6 and 7, we find that the best machine learning forecasting model is AdaBoost, followed by Random Forest, and K-nearest neighbors. The value of the errors looking at these three machine algorithms is very far from the errors we obtain with the other modelings. Looking at the errors when we compare the results using cryptocurrencies on the one hand, or adding stocks or commodities on the other side, for spot, we have a slight enhancement of the errors (when we add stocks or commodities). Looking at the MAPE statistic, Bitcoin is better explained by 'all' (0.31), followed by stocks (0.46), commodities (0.48), and finally cryptocurrencies (0.53).

Nevertheless, when we look at the futures price, the errors are higher for stocks and lower when adding commodities. The set of results is slightly different for futures: commodities return the lowest MAPE (0.34), followed by 'all' (0.41), cryptocurrencies (0.53), and finally stocks. The analysis shows a limited interest in adding commodities and financial assets to improve Bitcoin's forecasting power beyond that already captured by other cryptocurrencies.

If we compare with the results obtained in Table 3, for instance, MAPE = 0.11 to MAPE = 0.31 (Table 7), Tether does not improve the forecasting accuracy of the Bitcoin spot price. For the futures, we compare MAPE = 0.41 (Table 8) with 0.73 (Table 4): in that case, the conclusion is reverse. The question remains open whether this specific set of results is an artifact attributable to the stable coin Tether. Although Tether has a stable coin status, it is owned by the opaque (primarily unregulated) marketplace by BitMEX. It is not easy to assess the impact of that cryptocurrency on others. As Alexander et al. [117]

put it, regulators (need to) prioritize the investigation of the legitimacy of BitMEX and its contracts based on concerns of lack of transparency and potential market manipulation.

**Table 6.** Sub-sample with the Introduction of Tether (USDT): forecasting of the Coinbase Bitcoin Spot Price from 12 April, 2017, to 31 December, 2020.

| CRYPTO → SPOT BTC | AR(1) | ann | random | svm | knn | boost | ridge |
|---|---|---|---|---|---|---|---|
| RMSE | 3358.14 | 2582.10 | 296.02 | 2655.41 | 451.34 | 45.54 | 1529.02 |
| MAE | 2536.31 | 2025.75 | 157.97 | 2103.93 | 261.27 | 21.22 | 1130.54 |
| MAPE | 50.10 | 47.97 | 2.26 | 45.31 | 3.60 | 0.53 | 19.38 |
| STOCKS / BONDS / FX | AR(1) | ann | random | svm | knn | boost | ridge |
| RMSE | 3358.14 | 2867.13 | 319.45 | 2225.41 | 774.69 | 49.49 | 1801.17 |
| MAE | 2536.31 | 2035.10 | 192.76 | 1633.83 | 481.59 | 20.01 | 1286.77 |
| MAPE | 50.10 | 42.21 | 2.91 | 26.09 | 6.99 | 0.48 | 21.47 |
| COMMO | AR(1) | ann | random | svm | knn | boost | ridge |
| RMSE | 3358.14 | 2745.22 | 262.66 | 1734.43 | 1002.35 | 50.90 | 1339.07 |
| MAE | 2536.31 | 1928.04 | 148.56 | 1240.58 | 540.81 | 19.64 | 1011.37 |
| MAPE | 50.10 | 36.58 | 2.22 | 21.29 | 8.13 | 0.46 | 18.56 |
| ALL | AR(1) | ann | random | svm | knn | boost | ridge |
| RMSE | 3358.14 | 1776.32 | 228.64 | 1674.18 | 641.75 | 31.83 | 770.54 |
| MAE | 2536.31 | 1284.93 | 132.58 | 1297.20 | 367.47 | 12.04 | 617.04 |
| MAPE | 50.10 | 23.04 | 1.85 | 24.04 | 5.41 | 0.31 | 12.36 |

Note: AR(1) stands for the autoregressive model of order one; ann for the Artificial Neural Network model; random for the Random forest model; svm for the Support Vector Machine model; knn for the k-Nearest neighbor model; boost for the Adaboost model; and ridge for the Ridge regression. In terms of forecast statistics, we resort to the Root Mean Square Error (RMSE), the Mean Absolute Error (MAE), and the Mean Absolute Percent Error (MAPE).

**Table 7.** Sub-sample with the Introduction of Tether (USDT): Forecasting of the CME Bitcoin Futures price.

| CRYPTO → FUT BTC | AR(1) | ann | random | svm | knn | boost | ridge |
|---|---|---|---|---|---|---|---|
| RMSE | 3512.26 | 3008.13 | 472.54 | 3330.54 | 830.46 | 57.91 | 2022.13 |
| MAE | 2787.25 | 2533.75 | 207.79 | 2772.72 | 314.74 | 23.50 | 1567.53 |
| MAPE | 86.68 | 78.49 | 3.79 | 93.93 | 5.71 | 0.53 | 36.73 |
| STOCKS / BONDS / FX | AR(1) | ann | random | svm | knn | boost | ridge |
| RMSE | 3512.26 | 3235.90 | 439.44 | 2959.64 | 976.71 | 59.26 | 2295.76 |
| MAE | 2787.25 | 2466.18 | 217.48 | 2041.73 | 527.01 | 26.70 | 1586.74 |
| MAPE | 86.68 | 74.46 | 4.79 | 55.09 | 10.32 | 0.70 | 34.53 |
| COMMO | AR(1) | ann | random | svm | knn | boost | ridge |
| RMSE | 3512.26 | 2922.43 | 354.82 | 3121.00 | 1239.65 | 46.20 | 1613.32 |
| MAE | 2787.25 | 2018.50 | 169.06 | 2299.30 | 556.34 | 16.29 | 1140.27 |
| MAPE | 86.68 | 71.33 | 3.43 | 89.71 | 14.79 | 0.34 | 26.40 |
| ALL | AR(1) | ann | random | svm | knn | boost | ridge |
| RMSE | 3512.26 | 2495.86 | 424.99 | 2382.42 | 762.66 | 54.96 | 1440.18 |
| MAE | 2787.25 | 1984.92 | 182.68 | 1785.06 | 349.76 | 21.40 | 960.85 |
| MAPE | 86.68 | 55.38 | 5.36 | 55.33 | 6.35 | 0.41 | 22.73 |

Note: AR(1) stands for the autoregressive model of order one; ann for the Artificial Neural Network model; random for the Random forest model; svm for the Support Vector Machine model; knn for the k-Nearest neighbor model; boost for the Adaboost model; and ridge for the Ridge regression. In terms of forecast statistics, we resort to the Root Mean Square Error (RMSE), the Mean Absolute Error (MAE), and the Mean Absolute Percent Error (MAPE).

### 6.3. 2016–18 Bitcoin's Economic Cycle

The Bitcoin spot price was equal to 367 US$ in January 2016, skyrocketing to 19,891$ in December 2017, crashing to 3763$ in December 2018, for finally maintaining an apparent trend around 8000$ since 2019. This canonical decomposition into an economic cycle's phases of expansion and recession leads us to gauge the sensitivity of our results during the subperiod 1 January 2016–31 December 2018.

Regarding the Tables 8 and 9, results are similar—from the algorithm's race—to the full-sample predictions done previously, as AdaBoost is still the winner of horse race among competing machine learning models in all cases (followed by Random forest and kNN algorithms). Using only the information from other cryptocurrencies returns, we obtain the lowest forecast errors, both for spot (MAPE = 0.82) and futures (MAPE = 0.98) Bitcoin. By looking at the Adaboost spot results, for instance, we cannot assimilate Bitcoin to either financial securities (MAPE = 1.42 > 0.82) or commodities (MAPE = 1.07 > 0.82) because forecast errors are increasing. We notice that the errors are nearly the same when we look at the results for the futures. In conclusion, forecast errors are higher than during the whole sample. It can be linked to the explosiveness behavior of the Bitcoin price in December 2017.

**Table 8.** Sub-sample forecasting results of Coinbase Bitcoin Spot price during the 2016–18 economic cycle.

| CRYPTO → SPOT BTC | AR(1) | ann | random | svm | knn | boost | ridge |
|---|---|---|---|---|---|---|---|
| RMSE | 3347.17 | 2247.75 | 198.81 | 2182.41 | 331.18 | 22.50 | 701.02 |
| MAE | 2525.98 | 1888.10 | 79.46 | 1586.54 | 136.24 | 9.38 | 474.80 |
| MAPE | 165.32 | 141.22 | 2.32 | 85.90 | 3.31 | 0.82 | 29.41 |
| STOCKS / BONDS / FX | AR(1) | ann | random | svm | knn | boost | ridge |
| RMSE | 3347.17 | 2331.98 | 226.00 | 2161.75 | 556.07 | 31.85 | 1544.24 |
| MAE | 2525.98 | 1474.18 | 103.14 | 1515.32 | 252.90 | 14.69 | 1108.46 |
| MAPE | 165.32 | 79.01 | 3.53 | 107.53 | 5.83 | 1.42 | 75.59 |
| COMMO | AR(1) | ann | random | svm | knn | boost | ridge |
| RMSE | 3347.17 | 2456.96 | 239.60 | 1996.15 | 744.25 | 29.21 | 1350.50 |
| MAE | 2525.98 | 1391.29 | 101.55 | 1417.36 | 304.57 | 12.55 | 1000.99 |
| MAPE | 165.32 | 77.35 | 3.47 | 122.91 | 8.60 | 1.07 | 77.55 |
| ALL | AR(1) | ann | random | svm | knn | boost | ridge |
| RMSE | 3347.17 | 1592.95 | 185.40 | 2086.46 | 472.07 | 17.61 | 524.18 |
| MAE | 2525.98 | 1016.83 | 69.02 | 1567.12 | 207.21 | 7.78 | 360.27 |
| MAPE | 165.32 | 57.03 | 1.90 | 137.19 | 5.02 | 0.89 | 21.66 |

Note: AR(1) stands for the autoregressive model of order one; ann for the Artificial Neural Network model; random for the Random forest model; svm for the Support Vector Machine model; knn for the k-Nearest neighbor model; boost for the Adaboost model; and ridge for the Ridge regression. In terms of forecast statistics, we resort to the Root Mean Square Error (RMSE), the Mean Absolute Error (MAE), and the Mean Absolute Percent Error (MAPE).

**Table 9.** Sub-sample forecasting results of CME Bitcoin Futures price during the 2016–18 economic cycle.

| CRYPTO → FUT BTC | AR(1) | ann | random | svm | knn | boost | ridge |
|---|---|---|---|---|---|---|---|
| RMSE | 4363.09 | 3614.50 | 631.89 | 3802.03 | 1255.64 | 116.83 | 2006.04 |
| MAE | 3239.82 | 2960.70 | 298.57 | 3329.58 | 528.87 | 52.08 | 1408.66 |
| MAPE | 121.15 | 101.87 | 8.03 | 121.45 | 12.20 | 0.98 | 39.79 |
| STOCKS / BONDS / FX | AR(1) | ann | random | svm | knn | boost | ridge |
| RMSE | 4363.09 | 3900.97 | 711.14 | 3518.95 | 1208.71 | 98.46 | 3284.64 |
| MAE | 3239.82 | 2781.09 | 319.38 | 2384.01 | 550.15 | 45.42 | 2252.96 |
| MAPE | 121.15 | 81.72 | 6.93 | 53.42 | 10.85 | 1.05 | 49.91 |
| COMMO | AR(1) | ann | random | svm | knn | boost | ridge |
| RMSE | 4363.09 | 3749.76 | 650.19 | 3523.16 | 1763.10 | 89.78 | 2379.86 |
| MAE | 3239.82 | 2713.36 | 294.69 | 2806.55 | 855.68 | 38.94 | 1577.59 |
| MAPE | 121.15 | 72.63 | 6.43 | 106.25 | 21.38 | 0.98 | 43.29 |
| ALL | AR(1) | ann | random | svm | knn | boost | ridge |
| RMSE | 4363.09 | 2864.04 | 564.27 | 2954.87 | 1027.13 | 96.35 | 1612.76 |
| MAE | 3239.82 | 1911.06 | 188.29 | 2309.58 | 439.60 | 44.54 | 1123.03 |
| MAPE | 121.15 | 46.05 | 3.30 | 77.10 | 8.25 | 1.02 | 31.65 |

Note: AR(1) stands for the autoregressive model of order one; ann for the Artificial Neural Network model; random for the Random forest model; svm for the Support Vector Machine model; knn for the k-Nearest neighbor model; boost for the Adaboost model; and ridge for the Ridge regression. In terms of forecast statistics, we resort to the Root Mean Square Error (RMSE), the Mean Absolute Error (MAE), and the Mean Absolute Percent Error (MAPE).

*6.4. Year 2019*

We narrow down our analysis to the latest trading year in our dataset, a.k.a the year 2019, which did not bring bulls run or rallies in the price path of Bitcoin, which stayed at around 8000$. It is also far away from the next halving period, which should occur in May 2020 at current hash rates. The mining premium is halved for every 210,000 transaction blocks. About 50 bitcoins were generated every 10 min or so during the first four years, this value increased to 25 Bitcoin on 28 November 2012, and to 12.5 Bitcoin on 9 July 2016.

Sub-Sample Results

According to Tables 10 and 11, the prediction of Bitcoin is best based on Adaboost and Random Forest. SVM performs poorly. Robustness checks validate the main forecasting results and the hypothesis of Bitcoin segmentation within cryptocurrencies. The lowest MAPE for spot forecasts is achieved for all series (=0.18). The lowest MAPE for futures forecasts is achieved for commodities (=0.18). The results are globally similar to the full-period results (2015–2020). Thus, it appears some stability during 2019 for forecasting the Bitcoin spot price, whose errors are close to the errors obtained during the whole period (2015–2020).

For the futures price, it is worth noting that the evolution of the futures market during 2019 yields increased maturity and liquidity compared to the year 2018 (which can be seen as a trial period). During the 2019 subsample, we have a better view of the futures trading activity, given its overall stability and being free of the explosiveness behavior compared to the full period estimates. During the initial 2018 year, shocks revealed the Bitcoin futures market's youth market in search of price support trends.

**Table 10.** Sub-sample forecasting results of Coinbase Bitcoin Spot price during the year 2019.

| CRYPTO → SPOT BTC | AR(1) | ann | random | svm | knn | boost | ridge |
|---|---|---|---|---|---|---|---|
| RMSE | 2004.38 | 1839.90 | 269.78 | 1242.31 | 482.95 | 56.39 | 948.21 |
| MAE | 1621.48 | 1480.75 | 188.88 | 944.23 | 334.25 | 22.33 | 753.79 |
| MAPE | 19.15 | 19.03 | 2.20 | 11.47 | 3.77 | 0.31 | 8.99 |
| STOCKS / BONDS / FX | AR(1) | ann | random | svm | knn | boost | ridge |
| RMSE | 2004.38 | 1161.05 | 207.41 | 789.85 | 658.27 | 52.20 | 796.62 |
| MAE | 1621.48 | 960.84 | 146.26 | 658.62 | 494.73 | 23.82 | 641.54 |
| MAPE | 19.15 | 12.09 | 1.66 | 7.71 | 5.58 | 0.32 | 7.65 |
| COMMO | AR(1) | ann | random | svm | knn | boost | ridge |
| RMSE | 2004.38 | 1842.40 | 301.12 | 951.06 | 565.41 | 48.58 | 572.25 |
| MAE | 1621.48 | 1501.78 | 154.87 | 778.14 | 415.79 | 22.93 | 449.21 |
| MAPE | 19.15 | 19.21 | 1.72 | 9.65 | 4.80 | 0.30 | 5.51 |
| ALL | AR(1) | ann | random | svm | knn | boost | ridge |
| RMSE | 2004.38 | 1680.12 | 207.25 | 893.35 | 458.88 | 32.68 | 427.80 |
| MAE | 1621.48 | 1320.86 | 140.69 | 707.09 | 317.84 | 13.14 | 328.80 |
| MAPE | 19.15 | 17.21 | 1.59 | 8.52 | 3.60 | 0.18 | 3.86 |

Note: AR(1) stands for the autoregressive model of order one; ann for the Artificial Neural Network model; random for the Random forest model; svm for the Support Vector Machine model; knn for the k-Nearest neighbor model; boost for the Adaboost model; and ridge for the Ridge regression. In terms of forecast statistics, we resort to the Root Mean Square Error (RMSE), the Mean Absolute Error (MAE), and the Mean Absolute Percent Error (MAPE).

**Table 11.** Sub-sample forecasting results of CME Bitcoin Futures price during the year 2019.

| CRYPTO → FUT BTC | AR(1) | ann | random | svm | knn | boost | ridge |
|---|---|---|---|---|---|---|---|
| **RMSE** | 2526.17 | 2274.59 | 336.84 | 2274.13 | 774.28 | 51.70 | 859.54 |
| **MAE** | 2161.69 | 1844.83 | 216.16 | 1843.04 | 400.58 | 19.00 | 665.93 |
| **MAPE** | 28.19 | 27.87 | 2.85 | 27.79 | 4.76 | 0.28 | 8.54 |
| **STOCKS / BONDS / FX** | AR(1) | ann | random | svm | knn | boost | ridge |
| **RMSE** | 2526.17 | 2261.82 | 262.58 | 1486.93 | 870.11 | 39.54 | 1141.02 |
| **MAE** | 2161.69 | 1812.25 | 156.25 | 1133.90 | 627.91 | 12.06 | 889.87 |
| **MAPE** | 28.19 | 27.70 | 1.93 | 14.25 | 7.89 | 0.20 | 10.98 |
| **COMMO** | AR(1) | ann | random | svm | knn | boost | ridge |
| **RMSE** | 2526.17 | 2246.32 | 240.88 | 1699.47 | 784.03 | 34.63 | 837.17 |
| **MAE** | 2161.69 | 1847.42 | 142.97 | 1360.78 | 426.04 | 11.74 | 663.65 |
| **MAPE** | 28.19 | 28.10 | 1.63 | 19.23 | 5.25 | 0.18 | 8.62 |
| **ALL** | AR(1) | ann | random | svm | knn | boost | ridge |
| **RMSE** | 2526.17 | 2112.05 | 232.87 | 1844.38 | 507.67 | 38.29 | 598.87 |
| **MAE** | 2161.69 | 1753.21 | 150.59 | 1428.00 | 338.29 | 16.17 | 463.48 |
| **MAPE** | 28.19 | 26.91 | 1.82 | 20.40 | 4.16 | 0.23 | 6.07 |

Note: AR(1) stands for the autoregressive model of order one; ann for the Artificial Neural Network model; random for the Random forest model; svm for the Support Vector Machine model; knn for the k-Nearest neighbor model; boost for the Adaboost model; and ridge for the Ridge regression. In terms of forecast statistics, we resort to the Root Mean Square Error (RMSE), the Mean Absolute Error (MAE), and the Mean Absolute Percent Error (MAPE).

*6.5. 2020: The Next "Bull Run"?*

Market observers have suggested that the year 2020 would be bound to new "all-time highs", partly due to the halving of Bitcoin mining rewards, partly because of investors' behavior (FOMO, or fear-of-missing-out). Bill Gates advised against such investment, by stating merely that "*Bitcoin will randomly go up or down, so you should probably watch out*". (See MSN Money (2021) at https://www.msn.com/en-us/money/companies/bill-gates-vs-elon-musk-over-bitcoin/ar-BB1e1p7v, accessed on 2 February 2021). Many shrewd investors in the vein of Bill Gates have similarly noticed the "Tulip Mania" around Bitcoin in 2020, which led in the 1600s to the first recorded story of a financial bubble. Or maybe the blockchain revolution is inevitable, much like the internet revolution for its contemporary back in 1994 (when modem connection to the world wide web was in its infancy)? Only History will tell.

In parallel, Bitcoin has been advanced as a refuge for money during the Covid-19 sanitary crisis, challenging the role of Gold for several years to come. With the arrival of new institutional investors (who pledged, for instance, 5% of their portfolio allocation to Bitcoin futures (Kraken [119] documents BTC investments coming from several institutional investors, such as JP Morgan, Massachusetts Mutual Life Insurance, One River Asset Management, Guggenheim Global, Jefferies Investment Bank in the US, or BBVA in Switzerland)), a new economic cycle seems to have begun breaking all previous lines of resistance (that of $30,000 significantly) for BTC traders with increased market liquidity ($126 billion worth of trading in Bitcoin in December 2020, with a record-high of $16 billion traded on December 30 alone). The interest in Bitcoin as a financial store of value is further confirmed by the recent interest of hedge funds, such as Black Rock, who has begun entering the Bitcoin space (without revealing precisely the percentage of exposure to Bitcoin in its portfolio). (See CNBC (2021) at https://www.cnbc.com/2021/02/17/blackrock-has-started-to-dabble-in-bitcoin-says-rick-rieder.html, accessed on 17 February 2021).

The main findings from Tables 12 and 13 can be summarized as follows. In 2020, the quality of forecasts was overall the same as in 2019, with the Adaboost model standing out as the best machine learning model. The kNN, random forest, or Stochastic Gradient Descent algorithms rank closely as second best models depending on the statistic used. Nonetheless, we remark a higher dispersion across the statistics used for prediction, especially for the spot. This may be linked to the parabolic rise near the end of 2020.

**Table 12.** Sub-sample forecasting results of Coinbase Bitcoin Spot price during the year 2020.

| CRYPTO → SPOT BTC | AR(1) | ann | random | svm | knn | boost | ridge |
|---|---|---|---|---|---|---|---|
| **RMSE** | 1414.49 | 3307.80 | 197.38 | 2900.38 | 349.39 | 33.70 | 338.91 |
| **MAE** | 1070.73 | 2459.54 | 103.34 | 2052.99 | 234.06 | 17.80 | 241.83 |
| **MAPE** | 11.22 | 24.93 | 0.84 | 17.58 | 2.25 | 0.20 | 2.39 |
| **STOCKS / BONDS / FX** | AR(1) | ann | random | svm | knn | boost | ridge |
| **RMSE** | 1414.49 | 2512.15 | 311.51 | 2529.41 | 740.23 | 78.17 | 1471.50 |
| **MAE** | 1070.73 | 1975.08 | 180.96 | 1894.31 | 459.15 | 38.99 | 1163.68 |
| **MAPE** | 11.22 | 18.58 | 1.71 | 16.96 | 4.23 | 0.41 | 11.20 |
| **COMMO** | AR(1) | ann | random | svm | knn | boost | ridge |
| **RMSE** | 1414.49 | 2883.19 | 357.88 | 2420.39 | 676.30 | 96.19 | 1304.25 |
| **MAE** | 1070.73 | 2249.76 | 184.10 | 1728.99 | 417.52 | 46.36 | 996.39 |
| **MAPE** | 11.22 | 22.40 | 1.76 | 14.71 | 4.09 | 0.47 | 9.49 |
| **ALL** | AR(1) | ann | random | svm | knn | boost | ridge |
| **RMSE** | 1414.49 | 1644.56 | 196.82 | 2632.90 | 358.02 | 44.44 | 262.89 |
| **MAE** | 1070.73 | 1382.10 | 101.25 | 1860.75 | 249.53 | 22.85 | 187.06 |
| **MAPE** | 11.22 | 13.66 | 0.85 | 15.90 | 2.41 | 0.25 | 1.80 |

Note: AR(1) stands for the autoregressive model of order one; ann for the Artificial Neural Network model; random for the Random forest model; svm for the Support Vector Machine model; knn for the k-Nearest neighbor model; boost for the Adaboost model; and ridge for the Ridge regression. In terms of forecast statistics, we resort to the Root Mean Square Error (RMSE), the Mean Absolute Error (MAE), and the Mean Absolute Percent Error (MAPE).

**Table 13.** Sub-sample forecasting results of CME Bitcoin Futures price during the year 2020.

| CRYPTO → FUT BTC | AR(1) | ann | random | svm | knn | boost | ridge |
|---|---|---|---|---|---|---|---|
| **RMSE** | 1410.81 | 3312.17 | 215.78 | 3393.32 | 707.28 | 58.42 | 534.98 |
| **MAE** | 1067.87 | 2445.28 | 134.71 | 2490.83 | 306.53 | 26.25 | 385.49 |
| **MAPE** | 11.01 | 24.57 | 1.28 | 24.85 | 2.89 | 0.27 | 3.77 |
| **STOCKS / BONDS / FX** | AR(1) | ann | random | svm | knn | boost | ridge |
| **RMSE** | 1410.81 | 1962.39 | 324.59 | 2734.47 | 771.51 | 46.64 | 2074.45 |
| **MAE** | 1067.87 | 1486.87 | 167.88 | 1937.63 | 465.15 | 17.82 | 1598.52 |
| **MAPE** | 11.01 | 14.34 | 1.46 | 17.68 | 4.25 | 0.18 | 15.33 |
| **COMMO** | AR(1) | ann | random | svm | knn | boost | ridge |
| **RMSE** | 1410.81 | 2888.13 | 455.97 | 2350.38 | 709.81 | 60.48 | 1396.80 |
| **MAE** | 1067.87 | 2251.41 | 223.21 | 1670.00 | 429.37 | 22.26 | 1105.12 |
| **MAPE** | 11.01 | 22.37 | 2.14 | 15.51 | 4.04 | 0.22 | 10.72 |
| **ALL** | AR(1) | ann | random | svm | knn | boost | ridge |
| **RMSE** | 1410.81 | 1520.44 | 401.66 | 1910.55 | 352.58 | 53.12 | 659.22 |
| **MAE** | 1067.87 | 1217.22 | 148.97 | 1268.20 | 248.24 | 20.27 | 515.52 |
| **MAPE** | 11.01 | 11.96 | 1.44 | 12.25 | 2.32 | 0.20 | 5.43 |

Note: AR(1) stands for the autoregressive model of order one; ann for the Artificial Neural Network model; random for the Random forest model; svm for the Support Vector Machine model; knn for the k-Nearest neighbor model; boost for the Adaboost model; and ridge for the Ridge regression. In terms of forecast statistics, we resort to the Root Mean Square Error (RMSE), the Mean Absolute Error (MAE), and the Mean Absolute Percent Error (MAPE).

Despite the soaring prices near the end of the year 2020, ultimately breaching the barrier of $30k at the beginning of 2021, it is reassuring to observe that the machine learning models implemented keep delivering approximately the same forecasting accuracy metric as when the prices were in the low $3k or $8k average. We are confident that, for instance, the Adaboost or Random forest algorithms could be implemented as a decision-making tool in a banking environment, either for spot or futures forecasts, given their stability.

*6.6. Trading Strategies*

Most of the crypto traders continue to use chartist methods to decide to sell or buy (Shynkevich [120]). They also use the notion of profit and loss. With our exercise, we can provide interesting information to these investors who are interested in short positions.

Indeed, with the trends we observe in our predictions, we can provide new alternative trading strategies.

### 6.6.1. Hit Rates

To achieve this objective, we follow the methodology by [64]. We assess the accuracy of the Bitcoin price predictions based on the following formula:

$$\text{Hit rate} = \frac{h}{n}, \tag{16}$$

with $h$ the number of out-of-sample correct forecasts of the Bitcoin underlying price (e.g., spot or futures), and $n$ the number of tests (a.k.a, sessions of tests of different sizes depending on full- or sub-samples).

The key idea behind computing hit rates is to assess whether the ML algorithms can help the trader in his daily routine (by giving him consistent up/down price directions), or whether the trader is better off counting on his luck, a.k.a, flipping a coin every morning with 50% of success whatsoever (independent trials).

Regarding Table 14, we report a superior forecasting accuracy superior to 50% chance of flipping a coin. Ada Boost performs best above 70% in all cases in the full sample. Performance is consistent across the various subsamples. Variations exist, as the lowest hit rate is recorded at 61.89% for Bitcoin spot during the year 2019, whereas the highest hit rate is equal to 93.90% for Bitcoin spot during the 2016–18 economic cycle for all series. Random forest ranks second, oscillating around 60% for the full sample. The hit rate of Random forest is even higher for futures Bitcoin, where it can beat in one case the Ada Boost model (for cryptocurrencies only; random = 73.17% whereas boosting = 70.43%). For the subsamples, the Random forest model's predictive power remains around the hit rate of 60%. Its performance is slightly better when the underlying asset is the futures (see especially the case of the Bitcoin 2016–18 economic cycle sub-sample, where the Random forest ranks first in three cases). *k*NN and ridge regression hit consistently above than 50% threshold. AR(1), ANN, and SVM perform poorly (below 50% chance of flipping a coin). In 2020, the best hit rates ranged around 62%, down from 65% during the year 2019 (maybe attributable to the intramonth volatility push during December 2020?). Besides, we identify two instances (for spot forecasts) during which the random forest algorithm beats the Adaboost one.

In the literature, other authors have also studied the interest of using machine learning to predict asset prices. In Atsalakis et al. [64], the PATSOS (neuro-fuzzy algorithm) hit rate to predict Bitcoin is equal to 63.22%. In our approach, we perform slightly better for Bitcoin spot and futures, on specific periods. In 2019, our performance was in the range of literature by Atsalakis et al. [64]. In Fischer and Krauss [121], the average accuracy of the random forest algorithm to predict the S&P 500 oscillates between 52% and 57%. Based on Random Forest and Gradient Descent, Saad et al. [122] evaluate that a state-of-the-art machine learning model can achieve more than 90% forecast accuracy of the Bitcoin price (based on MAE and RMSE statistics).

Regarding the prediction of cryptocurrency returns, [123] document that machine learning classification algorithms reach about 55-65% predictive accuracy on average at the daily or minute level frequencies. However, our results depart from the latter authors. They attributed the best and consistent results in terms of predictive accuracy to the support vector machines compared to the logistic regression, artificial neural networks, and random forest classification algorithms.

**Table 14.** Hit rate performance comparison.

| Full sample/Spot | AR(1) | ann | random | svm | knn | boost | ridge |
|---|---|---|---|---|---|---|---|
| Hit rate (%) CRYPTO → SPOT BTC | <50% | <50% | 61.59% | <50% | 52.74% | 83.84% | 51.83% |
| Hit rate (%) STOCKS / BONDS / FX | <50% | <50% | 63.72% | 50.30% | 51.83% | 83.54% | 50.61% |
| Hit rate (%) COMMO | <50% | <50% | 60.06% | <50% | 53.35% | 85.37% | 51.22% |
| Hit rate (%) ALL | <50% | <50% | 57.93% | <50% | 51.83% | 88.41% | 51.83% |
| Full sample/Futures | AR(1) | ann | random | svm | knn | boost | ridge |
| Hit rate (%) CRYPTO → FUT BTC | <50% | <50% | 73.17% | 50.91% | 57.32% | 70.43% | 50.91% |
| Hit rate (%) STOCKS / BONDS / FX | <50% | <50% | 66.16% | <50% | 56.40% | 82.32% | 51.22% |
| Hit rate (%) COMMO | <50% | <50% | 67.07% | 50.30% | 52.74% | 80.79% | 50.91% |
| Hit rate (%) ALL | <50% | <50% | 70.12% | 50.30% | 57.93% | 77.74% | 50.30% |
| Sub sample/Crypto-select/Spot | AR(1) | ann | random | svm | knn | boost | ridge |
| Hit rate (%) CRYPTO → SPOT BTC | <50% | <50% | 58.84% | 50.30% | 55.18% | 85.37% | 50.30% |
| Sub sample/Crypto-select/Futures | AR(1) | ann | random | svm | knn | boost | ridge |
| Hit rate (%) CRYPTO → FUT BTC | <50% | 50.00% | 62.50% | 50.91% | 52.74% | 83.23% | 50.61% |
| Sub sample/Tether/Spot | AR(1) | ann | random | svm | knn | boost | ridge |
| Hit rate (%) CRYPTO → SPOT BTC | <50% | <50% | 60.98% | 50.30% | 54.27% | 84.15% | 50.30% |
| Hit rate (%) STOCKS / BONDS / FX | <50% | <50% | 58.54% | 50.30% | 53.35% | 86.89% | 50.91% |
| Hit rate (%) COMMO | <50% | <50% | 61.59% | 50.61% | 52.74% | 84.15% | 50.91% |
| Hit rate (%) ALL | <50% | <50% | 58.23% | <50% | 53.05% | 87.80% | 50.91% |
| Sub sample/Tether/Futures | AR(1) | ann | random | svm | knn | boost | ridge |
| Hit rate (%) CRYPTO → FUT BTC | <50% | <50% | 66.46% | <50% | 54.88% | 78.05% | 50.61% |
| Hit rate (%) STOCKS / BONDS / FX | <50% | <50% | 64.33% | 51.22% | 57.32% | 76.52% | 50.91% |
| Hit rate (%) COMMO | <50% | <50% | 63.41% | 50.30% | 54.88% | 81.10% | 50.30% |
| Hit rate (%) ALL | <50% | <50% | 64.02% | 50.30% | 55.18% | 79.88% | 51.22% |
| Sub sample/BTC cycle/Spot | AR(1) | ann | random | svm | knn | boost | ridge |
| Hit rate (%) CRYPTO → SPOT BTC | <50% | <50% | 58.54% | 50.30% | 52.74% | 87.50% | 50.91% |
| Hit rate (%) STOCKS / BONDS / FX | <50% | <50% | 56.40% | <50% | 52.13% | 91.16% | 50.30% |
| Hit rate (%) COMMO | <50% | <50% | 58.84% | <50% | 51.83% | 89.33% | <50% |
| Hit rate (%) ALL | <50% | <50% | 54.88% | <50% | 50.91% | 93.90% | 50.30% |
| Sub sample/BTC cycle/Futures | AR(1) | ann | random | svm | knn | boost | ridge |
| Hit rate (%) CRYPTO → FUT BTC | <50% | <50% | 70.73% | 50.30% | 57.01% | 70.73% | 51.22% |
| Hit rate (%) STOCKS / BONDS / FX | <50% | <50% | 72.87% | <50% | 54.27% | 72.26% | 51.22% |
| Hit rate (%) COMMO | <50% | <50% | 66.46% | <50% | 54.57% | 77.74% | 51.22% |
| Hit rate (%) ALL | <50% | <50% | 74.09% | <50% | 56.10% | 69.51% | 51.52% |
| Sub sample/2019/Spot | AR(1) | ann | random | svm | knn | boost | ridge |
| Hit rate (%) CRYPTO → SPOT BTC | <50% | <50% | 58.54% | 50.30% | 53.66% | 61.89% | 50.91% |
| Hit rate (%) STOCKS / BONDS / FX | <50% | <50% | 55.49% | 50.91% | 52.13% | 65.55% | 51.22% |
| Hit rate (%) COMMO | <50% | <50% | 58.54% | <50% | 52.13% | 64.02% | 50.61% |
| Hit rate (%) ALL | <50% | <50% | 57.01% | 50.30% | 52.44% | 64.63% | 50.91% |
| Sub sample/2019/Futures | AR(1) | ann | random | svm | knn | boost | ridge |
| Hit rate (%) CRYPTO → FUT BTC | <50% | <50% | 56.40% | <50% | 51.83% | 66.77% | 50.30% |
| Hit rate (%) STOCKS / BONDS / FX | <50% | <50% | 54.57% | <50% | 52.44% | 68.60% | 50.30% |
| Hit rate (%) COMMO | <50% | <50% | 54.88% | <50% | 51.83% | 68.60% | <50% |
| Hit rate (%) ALL | <50% | <50% | 57.01% | <50% | 52.13% | 66.77% | <50% |
| Sub sample/2020/Spot | AR(1) | ann | random | svm | knn | boost | sgd |
| Hit rate (%) CRYPTO → SPOT BTC | <50% | <50% | 60.06% | 50.30% | 53.05% | 62.50% | 50.00% |
| Hit rate (%) STOCKS / BONDS / FX | <50% | <50% | 61.28% | 50.91% | 54.57% | 57.93% | 50.61% |
| Hit rate (%) COMMO | <50% | <50% | 66.46% | 50.91% | 53.05% | 53.66% | 51.22% |
| Hit rate (%) ALL | <50% | <50% | 59.76% | 50.30% | 51.83% | 60.67% | 52.74% |
| Sub sample/2020/Futures | AR(1) | ann | random | svm | knn | boost | sgd |
| Hit rate (%) CRYPTO → FUT BTC | <50% | <50% | 61.28% | <50% | 52.13% | 61.89% | <50% |
| Hit rate (%) STOCKS / BONDS / FX | <50% | <50% | 59.15% | <50% | 53.66% | 62.50% | 50.30% |
| Hit rate (%) COMMO | <50% | <50% | 58.54% | <50% | 52.13% | 64.94% | <50% |
| Hit rate (%) ALL | <50% | <50% | 60.06% | <50% | 51.83% | 61.59% | 51.83% |

Note: AR(1) stands for the autoregressive model of order one; ann for the Artificial Neural Network model; random for the Random forest model; svm for the Support Vector Machine model; knn for the k-Nearest neighbor model; boost for the Adaboost model; and ridge for the Ridge regression.

6.6.2. ML Trading Results Contrasted with HODL Strategy

HODL is a term derived from a misspelling of 'hold' that refers to buy-and-hold strategies in the context of Bitcoin and other cryptocurrencies. (It originated from a famous post on the internet forum 'bitcointalk': https://bitcointalk.org/index.php?topic=375643.0, accessed on 27 May 2021). In the Markowitz world, it is equivalent to the 'buy-and-hold' strategy, which will be used as our benchmark against which ML algorithms compete.

Over the five years (13 January 2015 to 31 December 2020), we set the training period $T_{train} = 4 - $ year and the trading period $T_{trade} = 1 - $ year. The investor buys 100,000$ worth of Bitcoin at the beginning of the trading session and sells it in the end. His gains are evaluated thanks to the following Rate of Return (RoR) formulation:

$$\text{RoR} = \frac{\text{net gain from Bitcoin}}{\text{initial investment}}. \tag{17}$$

The ML algorithms function as automated trading bots, with the following instructions: (i) if the price forecast is equal to the current price, do not take any action; (ii) if the price forecast is below the current price, trigger a sell signal; (iii) if the price forecast is above the current price, initiate a buy signal. By following these instructions, the trader can initiate automated trading (of course, by implementing additional stop-loss rules for his P&L). Transaction fees are ignored, as they are documented to be virtually non-existent. For instance, on 26 October 2020, two back-to-back transactions of 45,671 Bitcoin ($602 million) and 43,185 Bitcoin ($570 million), minutes apart, were sent from a Xapo wallet to two addresses. That corresponds to over $1.1 billion in Bitcoin. The total transaction fee of $3.54 spent was about the price of a Starbucks coffee. (See, e.g., https://decrypt.co/46346/someone-just-sent-1-billion-in-bitcoin-paid-only-3-in-fees, accessed on 26 October 2020. For real-time mean transaction fees for BTC, see Coinmetrics website: https://network-charts.coinmetrics.io/, accessed on 27 May 2021).

Table 15 compares the profit and loss account (P&L) from the HODL strategy with that of each machine learning algorithm (in the case where the trader follows their price signals) we use all along in this paper. As a trader, we expect a positive RoR and as high as possible.

We observe that the best RoR is achieved by the HODL strategy (26.88%). Looking at our results, depending on the data set used, the best RoR is not always given by the same algorithm. Depending on the dataset we use, the best RoR is provided by (i) an AR(1) for cryptocurrencies (25.55%), (ii) an AR(1) for stocks and bonds (25.55%), (iii) an ANN using commodities (26%), (iv) by Ridge using all the variables (30.47%). There is no segmentation since the best results are obtained as we add financial variables. The cryptocurrency world is not the best trading strategy. ANN using commodities, we have the same result as all (26–26.88%).

Our results stand in sharp contrast with Atsalakis et al. [64] (PATSOS neuro-fuzzy = 37.34%): although the ML algorithms deliver high forecast accuracy, they do not necessarily convert into money (candlestick patterns can be tricky to non-experienced investors). For sensitivity analysis, we run again the trading strategy with coins bought in January 2019 and sold at the end of our study period (aka, December 2020). The results are displayed in Table 16. The latest timeframe chosen tends to exacerbate the forecast statistics due to a heightened speculative situation on cryptocurrency markets, but it does not change the core finding of this section (in favor of the HODL strategy for Bitcoin versus market gains issued from machine learning models).

**Table 15.** Bitcoin Rate of Return (RoR) of ML algorithms against the HODL strategy.

| Strategy | Bitcoin RoR |
|---|---|
| HODL | 26.88% |
| Cryptos only | |
| AR(1) | 25.55% |
| ann | 12.26% |
| random | 18.93% |
| svm | 0.05% |
| knn | 19.00% |
| boost | 22.47% |
| ridge | 21.53% |
| Stocks/bonds/fx only | |
| AR(1) | 25.55% |
| ann | 5.13% |
| random | 21.50% |
| svm | 0.04% |
| knn | 17.48% |
| boost | 22.48% |
| ridge | 21.48% |
| Commodities only | |
| AR(1) | 25.55% |
| ann | 26.00% |
| random | 18.04% |
| svm | 0.07% |
| knn | 16.77% |
| boost | 22.48% |
| ridge | 13.00% |
| All | |
| AR(1) | 25.55% |
| ann | 15.20% |
| random | 22.75% |
| svm | 0.05% |
| knn | 24.23% |
| boost | 22.37% |
| ridge | 30.47% |

Note: The RoR is computed for an initial investment of 100,000$ during 13 January 2015–31 December 2020, with three-year initial training period for the ML algorithms, and two-years of paper trading before cashing out. AR(1) stands for the autoregressive model of order one; ann for the Artificial Neural Network model; random for the Random forest model; svm for the Support Vector Machine model; knn for the k-Nearest neighbor model; boost for the Adaboost model; and ridge for the Ridge regression.

Considering the different algorithms, the traditional econometrics AR(1) strategy fares surprisingly well in this context (best results for cryptocurrencies only, stocks/bonds/FX only). We note the stability of the AdaBoost and Random forest strictly after AR(1). ANN comes first (commodities only) due to finding better entry/exit prices in that particular case. Ridge regression performs best for all, being able to find as well an excellent combination of buying/selling points. Unlike other parts of our paper, we cannot say that one machine learning algorithm outperforms the others.

The philosophy of this last small exercise is that: (i) It is challenging to grab daily gains. Many investors report losses to the IRS for tax rebates ('Get rich or die tryin'). (ii) 'REKT': many investors are 'wrecked' by margin trading and leverage on BitMEX (reserved for shrewd FX traders). (iii) Machine Learning algorithms only give trading signals and do not teach how to trade. Traders resort to support and resistance lines, filters, technical indicators in TradingView (a list of trading rules such as Relative Strength Index (RSI), MACD curves, Exponential Moving Averages, Momentum, and their specifications can be

found in Shynkevich [120]). For a comprehensive examination of technical trading rules in cryptocurrency markets, see also [124].

**Table 16.** Sensitivity Analysis for Bitcoin Rate of Return (RoR) of ML algorithms against the HODL strategy.

| Strategy | Bitcoin RoR |
|:---:|:---:|
| HODL | 27.46% |
| Cryptos only | |
| AR(1) | 26.13% |
| ann | 12.79% |
| random | 19.55% |
| svm | 0.06% |
| knn | 19.61% |
| boost | 23.06% |
| ridge | 22.09% |
| Stocks/bonds/fx only | |
| AR(1) | 26.13% |
| ann | 5.68% |
| random | 21.98% |
| svm | 0.05% |
| knn | 18.06% |
| boost | 23.05% |
| ridge | 22.06% |
| Commodities only | |
| AR(1) | 26.13% |
| ann | 26.58% |
| random | 18.56% |
| svm | 0.65% |
| knn | 17.36% |
| boost | 23.05% |
| ridge | 13.61% |
| All | |
| AR(1) | 26.13% |
| ann | 15.81% |
| random | 23.32% |
| svm | 0.07% |
| knn | 24.81% |
| boost | 23% |
| ridge | 31.05% |

Note: The RoR is computed for an initial investment of 100,000 $ during 01 January 2019–31 December 2020, with three-year initial training period for the ML algorithms, and two-years of paper trading before cashing out. AR(1) stands for the autoregressive model of order one; ann for the Artificial Neural Network model; random for the Random forest model; svm for the Support Vector Machine model; knn for the k-Nearest neighbor model; boost for the Adaboost model; and ridge for the Ridge regression.

Bitcoin currently being in a bear market since the 19k$ price spike, our paper trading P&L challenges the thirst to be a millionaire ('When Lambo?') often found in the trading forums of the crypto-sphere, such as Reddit, where beginner day-traders dream of driving the most expensive sportscars—Lamborghinis.

## 7. Conclusions

Machine learning algorithms attempt to find natural patterns in data to enhance decision-making. Machine learning typically prescribes a vast collection of high-dimensional models attempting to predict quantities of interest to solve problems in computational finance while imposing regularization methods.

In a seminal paper, Zhao and Hastie [8] warns us against the pitfalls of 'black-box' machine learning models and urge us to make proper use of (i) a good predictive model, (ii) a sound selection of the dataset, and (iii) visualization tools to ensure the quality of the research work.

This paper addresses the following central research questions: How far the is use of machine learning techniques useful to forecast daily movements of the price of Bitcoin? How strong is the relationship between cryptocurrencies and traditional financial assets? How to perform a trading strategy based on cryptocurrencies? We overcome the risks of machine learning in predicting Bitcoin spot and futures prices by documenting: (i) either the AdaBoost or Random forest algorithms perform as the 'best' machine learning models among the six considered and could be implemented in a banking institution's internal computing system for Bitcoin forecastability. (ii) A financial market approach favors price relationships between asset classes (cryptocurrencies, stocks, bonds, foreign exchange, and commodities) for the dataset's quality. (iii) We implement a myriad of visualization tools (e.g., Louvain clustering, self-organizing map, t-distributed SNE, Sieve diagrams, and multidimensional scaling) to strengthen our findings.

During the period under study, the key takeaway is that Bitcoin appears as a problematic asset to forecast, subject to frequent price swings, as highlighted by the latest end-of-year 2020 run-up. From our empirical results, Bitcoin appears segmented from traditional financial and commodity markets. It seems to react more to the information content stemming from other cryptocurrencies, enhancing the forecast accuracy of Bitcoin. Whatever the period in which we work (full period, sub-sample periods), the best result is achieved during the full period (lowest MAPE = 0.15).

Across the trading strategies, we have documented that (i) machine learning algorithms (configured as bots following buy/sell signals) do not teach how to trade, (ii) the buy-and-hold strategy appears the best, which incites owners of Bitcoin to 'hodl'. Because of the variability of the forecasting results, it is necessary to let cool heads prevail before investing in Bitcoin with private individuals' money. Wealthy clients from investment banks can have access to Bitcoin funds in special entities such as Morgan Stanley, for individuals with "an aggressive risk tolerance" and a net worth of $2 million held by the firm. (See CNBC (2021) at https://www.cnbc.com/2021/03/17/bitcoin-morgan-stanley-is-the-first-big-us-bank-to-offer-wealthy-clients-access-to-bitcoin-funds.html, accessed on 17 March 2021). Such clients are especially waiting for the approval of ETF funds by the U.S. SEC led by the new appointment of Mr. Gary Gensler, a professor in the digital economy from MIT. The SEC's cryptocurrency commissioner, Ms. Hester Peirce (dubbed "crypto mom"), has long advocated a Bitcoin ETF. The argument behind the use of cryptocurrencies lies in the modernization of the financial system, whereby assets worth billion dollars can be transferred securely and quasi-instantly (compared to the SWIFT/IBAN system) between agents. Transaction fees are recorded to be minimal. It is possible to follow the transactions in real-time on `blockchair.com`. For example, one timestamp 6 April 2021 13:22 (UTC) records $1,783,170,000 exchanged through Bitcoin for a transaction fee of $45.45. At the time of writing, only Canada has approved the opening of ETFs on Bitcoin (such as Purpose Investment, Evolve Funds Group, or CI Galaxy). (See Coindesk (2021) at https://www.coindesk.com/third-bitcoin-etf-expected-to-launch-in-canada-this-week, accessed on 8 March 2021).

Overall, we believe that Bitcoin remains a difficult beast to tame for a modeler, and that the debate on its pricing and forecasting accuracy remains open. We wish to insist here on the speculative nature of investing in Bitcoin, which remains an artificial mechanism without an underlying (unless it can be considered that the proof-of-work algorithm is a receivable underlying in financial derivatives markets?). It remains hard to gauge the fundamental value of Bitcoin from an economist's standpoint, hence the challenge attached to any attempt at forecasting.

As avenues for future research, we would like to stress that our method can be extended to alt-coins, with a specific interest obviously for Ethereum, Litecoin, or Ripple. (i.e.,

the "main" alt-coins, as listed on https://coinmarketcap.com/, accessed on 27 May 2021). Having forecast the spot and futures prices of Bitcoin in this article, it might be interesting to look at the spreads between the two next. Indeed, with the "democratization" of ML among institutional investors, we can imagine that there will be more significant volumes of trading and arbitrage between the two markets, but the differences in performance between the two were due for a certain time. reflect distinct populations of participants.

**Author Contributions:** Conceptualization, J.C. and D.G.; methodology, J.C. and D.G.; software, J.C.; validation, J.C., D.G. and S.G.; data curation, J.C.; writing—original draft preparation, J.C. and D.G.; writing—review and editing, J.C., D.G. and S.G.; supervision, D.G.; project administration, J.C. All authors have read and agreed to the published version of the manuscript.

**Funding:** This research received no external funding.

**Institutional Review Board Statement:** Not applicable.

**Informed Consent Statement:** Not applicable.

**Data Availability Statement:** Data is publicly available on Coinbase: https://www.coinbase.com/fr/ accessed on 27 May 2021, Fred: https://fred.stlouisfed.org/ accessed on 27 May 2021.

**Acknowledgments:** For useful comments on earlier drafts, we thank Sofiane Aboura and Laurent Gauthier. The authors also wish to thank conference participants at the University of Geneva's MAF (Mathematical and Statistical Methods for Actuarial Sciences and Finance) in November 2020, at the 1st Research Day of the Macro-Finance Axis (Research Lab LED, University Paris 8) in April 2021, as well as at the RIFED Global Virtual Conference on Fintech, Business Ecosystem & Economic Development in June 2021.

**Conflicts of Interest:** The authors declare no conflict of interest.

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
