# Peer review of "Is It Possible to Forecast the Price of Bitcoin?"

_forecasting, doi:10.3390/forecast3020024_

Round 1

Reviewer 1 Report

Author try to predict BTC price based on machine learning algorithms. My main doubt is there were many trials to do this task more and less successful. E.g. Wołk, K. (2020). Advanced social media sentiment analysis for short‐term cryptocurrency price prediction. Expert Systems37(2), e12493. Jay, P., Kalariya, V., Parmar, P., Tanwar, S., Kumar, N., & Alazab, M. (2020). Stochastic neural networks for cryptocurrency price prediction. IEEE Access8, 82804-82818. Saad, M., Choi, J., Nyang, D., Kim, J., & Mohaisen, A. (2019). Toward characterizing blockchain-based cryptocurrencies for highly accurate predictions. IEEE Systems Journal14(1), 321-332. And much more. I would like to see in the text a deep analysis of current SOTA in this areas with different techniques, compared with each other and most importantly well described how author solutions differs from them and how improves SOTA. Some interesting articles are already mentioned but not all important ones and it not clear enough how authors solution differs. Will the method apply to alt-coins ? Why yes/not? It would be good extending research to other coins as well. Yes, bitcoin is most popular, but people gain most of their profits from altcoins. Do authors think that we can neglect social texts their sentiment, etc. ? E.g. Elon Muskt tweets can pump up or down BTC/DOGE price by a lot. How to predict such actions ? Finally is your method better than simply buy and hold strategy - Authors try to simulate for example if I bought coin in 2015 and sold in 2020. I think time period is very unfair…. It is from prom time period when marked was pumped to another pump. What would happen if I bought coins in 2019 and sold April 2021. Can you do such comparison ? It would be nice to compare long term investments vs short term. Any potential future plans or potential research avenues? I suggest adding in point short highlight section with main finding and differences.

Author Response

See pdf file attached

Reviewer 2 Report

The paper concerns a very relevant topic, is relatively well placed in the reference literature and is quite well written.

To achieve the bar of acceptance, it should be improved in a few directions

  1. Increase the literature review on machine learning methods to predict bitcoin and crypto prices. 
  2. Shorten the paper, to make it more readable. For example reducing the part on machine learning models, citing summary literature; or the introductory part on crypto assets
  3. There may be a "reverse causality" effect between the bitcoin price and that of the crypto assets. The authors should clarify how to deal with this aspect
  4. The use of RMSE to evaluate predictions is fine when the response is continuous but may not be so for a binary response such as "buy"/"sell". The authors should comment on this
  5. Machine learning models are well known to be black box and not explainable. The authors should comment on this and refer to recent papers in the field of explainable AI that use Shapley values or Lime to explain predictions, for example in credit risk/financial risk management
  6. The authors should comment better on the interconnectedness between the different crypto assets, and their high order interactions, using multivariate models such as Granger Causality (Billio et al 2012) Diebold and Yilmaz approaches for crypto assets and exchanges (JEC 2017 and further papers) and correlation networks (Abedifar et al., Journal of financial stability 2017; Ahelegbey et al., 2016, Journal of Applied Econometrics)
  7. The authors also consider the impact of news, as in the paper by Cerchiello and Nicola (2018, Entropy)

Author Response

See pdf file attached

Reviewer 3 Report

The paper's main focus was forecasting the price of a Bitcoin. The authors suggested y six machine learning2algorithms without assuming apriori their usefulness. It is a fascinating approach. Two things, however, need to be reformulated.

First, in my opinion, correlation analysis between Bitcoin and other asset classes is beyond the scope of the paper. The findings are significant but are not well aligned with the rest of the paper and the paper’s aim, which was related to predicting nonstationary time series. Second, authors should reconsider the description of their methods. The current description is written in a very encyclopedic form. It is hard to determine how each algorithm can contribute to the paper's central problem, which is both a long and short-term prediction of Bitcoin price. Furthermore, the authors should comment on a more general prediction of nonstationary time series via machine learning models.

Author Response

See pdf file attached

Round 2

Reviewer 1 Report

Authors complied to comments from 1st review and extended problematic parts of the article. Not 100% but in some cases thya provided justifications for their decisions. I suggest addid brief highlights sections before publication.

Reviewer 2 Report

The authors have satisfactorily taken my suggestions into account. The paper can be published

Reviewer 3 Report

The paper has been greatly improved and most of my comments have been included.